# Proteome profiling in cerebrospinal fluid reveals novel biomarkers of Alzheimer's disease

Jakob M Bader[1,†] [ID], Philipp E Geyer[1,2,†] [ID], Johannes B Müller[1] [ID], Maximilian T Strauss[1] [ID], Manja Koch[3] [ID], Frank Leypoldt[4,5], Peter Koertvelyessy[6,7], Daniel Bittner[6], Carola G Schipke[8], Enise I Incesoy[9], Oliver Peters[9,10], Nikolaus Deigendesch[11], Mikael Simons[12,13], Majken K Jensen[3,14], Henrik Zetterberg[15,16,17,18] & Matthias Mann[1,2,*] [ID]

## Abstract

Neurodegenerative diseases are a growing burden, and there is an urgent need for better biomarkers for diagnosis, prognosis, and treatment efficacy. Structural and functional brain alterations are reflected in the protein composition of cerebrospinal fluid (CSF). Alzheimer's disease (AD) patients have higher CSF levels of tau, but we lack knowledge of systems-wide changes of CSF protein levels that accompany AD. Here, we present a highly reproducible mass spectrometry (MS)-based proteomics workflow for the in-depth analysis of CSF from minimal sample amounts. From three independent studies (197 individuals), we characterize differences in proteins by AD status (> 1,000 proteins, CV < 20%). Proteins with previous links to neurodegeneration such as tau, SOD1, and PARK7 differed most strongly by AD status, providing strong positive controls for our approach. CSF proteome changes in Alzheimer's disease prove to be widespread and often correlated with tau concentrations. Our unbiased screen also reveals a consistent glycolytic signature across our cohorts and a recent study. Machine learning suggests clinical utility of this proteomic signature.

**Keywords** Alzheimer's disease; cerebrospinal fluid; mass spectrometry; neurodegeneration; proteomics

**Subject Categories** Biomarkers; Neuroscience; Proteomics
**Mol Syst Biol. (2020) 16: e9356**

## Introduction

Alzheimer's disease (AD) is the most common type of dementia, and its prevalence is growing rapidly in aging societies (GBD 2016 Neurology Collaborators, 2019). In 2015, almost 47 million people worldwide were estimated to be affected by dementia, and the numbers are expected to reach 75 million by 2030, and 131 million by 2050, with the greatest increase expected in low-income and middle-income countries (Winblad *et al*, 2016). Patients with AD typically present with memory impairment and difficulty performing activities of daily living (Scheltens *et al*, 2016). However, symptoms may manifest decades after the underlying pathology has initiated, including the deposition of amyloid plaques and development of neurofibrillary tangles (Jack *et al*, 2010).

Biomarkers have become important diagnostic tools to define the presence and absence of dementia before onset of memory loss. While a research framework for defining AD based on beta amyloid

1   Department of Proteomics and Signal Transduction, Max Planck Institute of Biochemistry, Martinsried, Germany
2   NNF Center for Protein Research, Faculty of Health Sciences, University of Copenhagen, Copenhagen, Denmark
3   Departments of Nutrition & Epidemiology, Harvard T.H. Chan School of Public Health, Boston, MA, USA
4   Institute of Clinical Chemistry, Faculty of Medicine, Kiel University, Kiel, Germany
5   Department of Neurology, Faculty of Medicine, Kiel University, Kiel, Germany
6   Department of Neurology, Medical Faculty, Otto von Guericke University Magdeburg, Magdeburg, Germany
7   Department of Neurology, Charité Universitätsmedizin Berlin, Berlin, Germany
8   Experimental & Clinical Research Center (ECRC), Charité – Universitätsmedizin Berlin, corporate member of Freie Universität Berlin, Humboldt-Universität zu Berlin, & Berlin Institute of Health, Berlin, Germany
9   Department of Psychiatry, corporate member of Freie Universität Berlin, Humboldt-Universität zu Berlin & Berlin Institute of Health, Charité Universitätsmedizin Berlin, Berlin, Germany
10  German Center for Neurodegenerative Diseases, Berlin, Germany
11  Institute of Medical Genetics and Pathology, University Hospital Basel, Basel, Switzerland
12  German Center for Neurodegenerative Diseases (DZNE), Munich, Germany
13  Munich Cluster for Systems Neurology, Munich, Germany
14  Department of Public Health, University of Copenhagen, Copenhagen, Denmark
15  Department of Psychiatry and Neurochemistry, Institute of Neuroscience and Physiology, the Sahlgrenska Academy at the University of Gothenburg, Mölndal, Sweden
16  Clinical Neurochemistry Laboratory, Sahlgrenska University Hospital, Mölndal, Sweden
17  UK Dementia Research Institute at UCL, London, UK
18  Department of Neurodegenerative Disease, UCL Institute of Neurology, London, UK
    *Corresponding author. Tel: +49 8985782557; E-mail: mmann@biochem.mpg.de
    †These authors contributed equally to this work

(Aβ) deposition, pathologic tau, and neurodegeneration (ATN) has been proposed (Jack *et al*, 2018), clinical criteria for AD are not universally standardized and range from clinical presentation to brain imaging by MRI and PET to clinical chemistry analysis of $A\beta_{1-42}/A\beta_{1-40}$, total-tau (t-tau), and phosphorylated-tau (p-tau$_{181}$) in cerebrospinal fluid (CSF; Frisoni *et al*, 2010; McKhann *et al*, 2011; Ferreira *et al*, 2014; Rice & Bisdas, 2017). Most research currently focuses on Aβ and tau, because they are the main components of amyloid plaques and neurofibrillary tangles (Serrano-Pozo *et al*, 2011). However, the search for a disease-modifying therapy has yet to show clinically relevant results and it is becoming increasingly clear that many additional pathological changes in multiple pathways occur in dementia.

Thus, we propose an unbiased analysis of CSF proteins in participants with and without AD for a comprehensive search for novel diagnostic biomarkers. A set of reliable protein biomarkers rather than a single marker could also enable the development of highly specific tests for early disease detection in at-risk segments of the population. Ideally, such markers should identify unexpected biological pathways and new potential therapeutic targets for future development.

Mass spectrometry (MS)-based proteomics has become a very powerful technology for the analysis of protein abundance levels, modifications, and interactions, with important discoveries in biological and biochemical research, including neuroscience (Aebersold & Mann, 2016; Hosp & Mann, 2017). MS-based proteomics is unbiased in the sense that it identifies and quantifies proteins in an untargeted manner. Additionally, the identification is extremely specific through the amino acid sequence information at the peptide level. These inherent features differentiate MS-based from affinity-based methods and should make MS an ideal tool for biomarker discovery; however, in body fluids this long-standing goal has not generally been realized so far. This has been due to a variety of technological and conceptual limitations, compromising reproducibility, the number of consistently quantified proteins and throughput (Geyer *et al*, 2017). For instance, a general issue in body fluid proteomics is the presence of highly abundant proteins such as albumin that hamper efficient identification of less abundant proteins. Previous workflows were laborious, typically quantified a few hundred proteins at most per sample and required hundreds of microliters of precious CSF, thereby limiting the availability of suitable samples (Dayon *et al*, 2018). Reproducibility was low with only a minority of proteins having clinically accepted coefficients of variation (CV) of < 20%. Furthermore, many proteins were not quantifiable in all study participants and validation in well-characterized study populations was lacking. Therefore, entire databases have been curated to navigate reported CSF proteome alterations across studies in the field of neurodegeneration including AD (Guldbrandsen *et al*, 2017).

Recent technological advances enable substantially higher proteome coverage and better and more comprehensive protein quantitation. These developments include automated sample preparation, technological improvements in mass spectrometers, MS data acquisition, and processing software that synergize to enhance the overall analytical performance (Bruderer *et al*, 2017; Kelstrup *et al*, 2018). Based on these advances, we here developed a streamlined and highly reproducible workflow from sample preparation to data-independent MS acquisition (Ludwig *et al*, 2018) and an integrated analysis of the results for CSF. This workflow enabled us to clearly identify the established markers as well as a large number of consistent and biologically meaningful proteome changes across several independent cohorts.

# Results

## Overview of study populations

We recently proposed a shift in the study design of clinical discovery proteomics termed "rectangular strategy" (Geyer *et al*, 2017). In the previous "triangular strategy" study design, selected samples were characterized with extensive workflows and a small number of candidates were then assessed in a larger number of individuals using targeted methods. However, these candidates often turned out to be specific to the discovery population and could not be validated in independent study populations. In contrast, in the "rectangular strategy", multiple studies are subjected to the same high proteome depth workflow, moving the discovery to the population-wide setting in order to discern pathological from study-specific effects.

To implement the rectangular strategy, we analyzed three separate study populations of about 30 AD patients and about 30 or 50 controls, amounting to 197 individuals in total (Fig 1A). We refer to the study populations as cohorts throughout the manuscript, because each cross-sectional study was slightly different, conducted in distinct settings and geographical regions. One cohort originated from western Sweden, another from the German cities Magdeburg and Kiel (obtained through Harvard T. H. Chan School of Public Health), and the third from Berlin. The overall median age was 70.0 ± 12.1 years (± SD) (Fig EV1A). However, the 16 non-AD control patients of the Kiel sub-cohort were younger (median 32.0 ± 17.1 years). In each of our cohorts, patients were classified as AD if the t-tau concentration was above 400 ng/l, and the $A\beta_{1-42}$ concentration below 550 ng/l or the $A\beta_{1-42}/A\beta_{1-40}$ ratio below 0.065 as determined by ELISA measurement at the clinical collection site (Materials and Methods).

The degree of separation of AD cases and controls by clinical AD CSF biomarker concentrations differed across cohorts. AD and non-AD were best separated in the Sweden cohort but the Magdeburg cohort also exhibited a good overall separation (Fig EV1B–K, Materials and Methods). In the Berlin cohort, however, AD and control groups overlapped to some degree regarding CSF $A\beta_{1-42}$ and slightly regarding t-tau.

## Characterization of the CSF proteomics workflow

Previously, we developed a streamlined Plasma Proteome Profiling pipeline, in which the proteins in one microliter of plasma are digested to peptides and purified for MS analysis in an automated system (Geyer *et al*, 2016). CSF contains much less protein than plasma, with about 0.17–0.70 g/l and 60–80 g/l total protein content, respectively (Seyfert *et al*, 2002; Laub *et al*, 2010). Nevertheless, we achieved a very robust workflow with high proteome depth from only a few microliter of sample that was not depleted of highly abundant proteins (Fig 1A and C). We adopted a data-independent acquisition strategy (DIA), both because it can achieve high

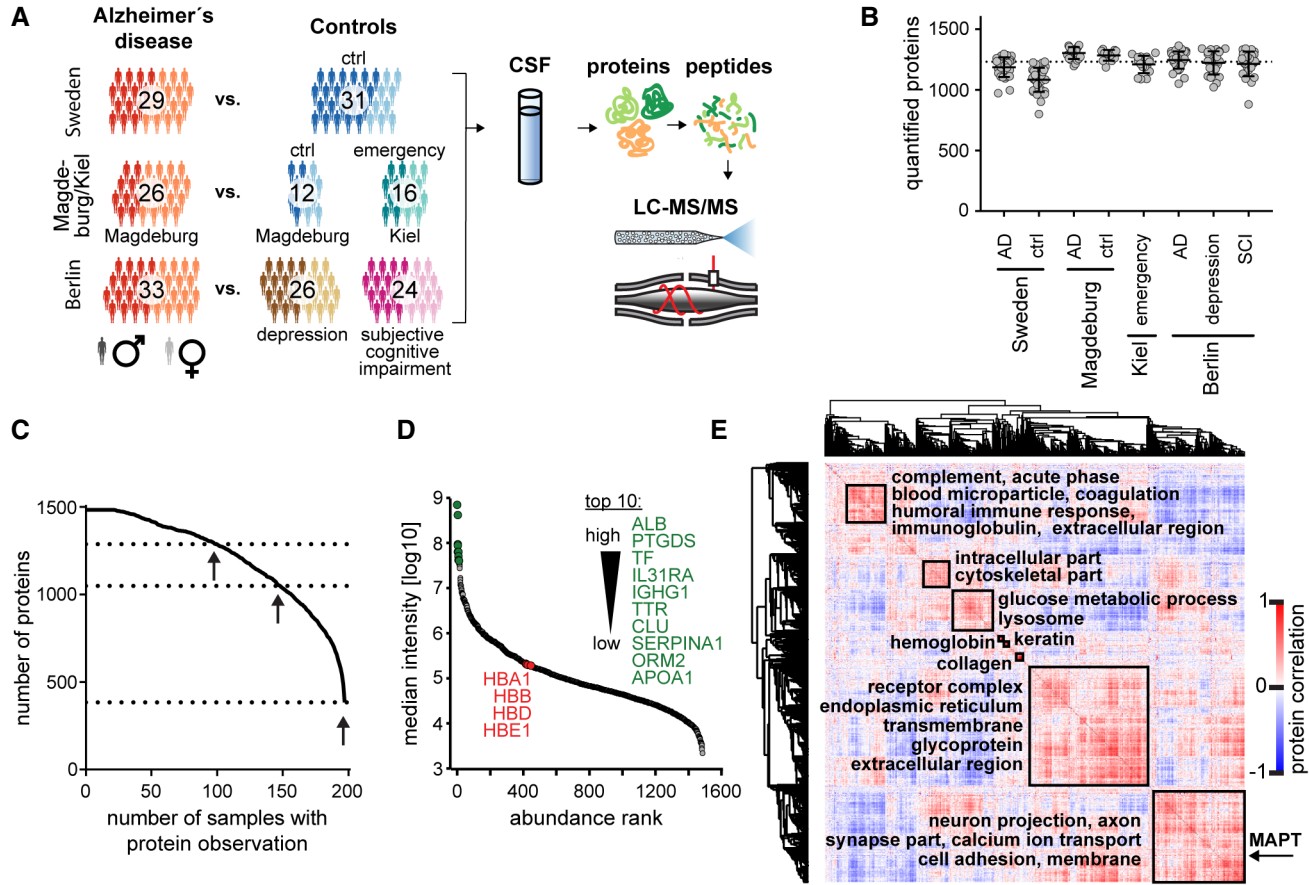

**Figure 1. Study overview and CSF proteome characterization.**

A  Overview of the study populations (cohorts) and schematic proteomic workflow. The CSF of three cohorts comprising AD and control subjects was analyzed. The total number of subjects per cohort group is depicted. Light and dark shades represent female and male subjects, respectively. "Ctrl" refers to non-AD control subjects.

B  Number of proteins identified and quantified passing the 1% FDR cutoffs in each sample. Horizontal lines show the mean and the error bars ± SD. The dashed line indicates the level of the meta-median (1,233 proteins) of the group medians of quantified proteins. Number of samples per group as shown in A).

C  Data completeness curve. The number of proteins in the dataset (Y axis) depending on the minimum number of samples in which the proteins have each been quantified (X axis) is plotted. The arrows indicate 50%, 75%, and 100% data completeness.

D  Median CSF protein abundance distribution as calculated from MS intensities of quantified peptides of each protein. The top ten most abundant proteins and hemoglobins are highlighted.

E  Global correlation map of proteins generated by clustering the Pearson correlation coefficients of all possible protein combinations. The abundance of proteins with common regulation correlates across samples, and they therefore form a cluster. Prominent clusters are annotated with functional terms obtained from bioinformatics enrichment analysis. The position of tau (gene name MAPT) is labeled on the Y axis. The inset shows the color code for Pearson correlation coefficients.

data completeness (Gillet *et al*, 2012) and because it has been shown to perform excellently on the linear quadrupole-Orbitrap instruments employed here (Kelstrup *et al*, 2018). A DIA library of about 2,700 proteins was computationally merged from pooled AD and non-AD samples after separation into 24 fractions each and a direct-DIA search for all single-run samples (Materials and Methods). CSF proteomes were acquired by measuring single 100-min gradient runs for each patient.

On average, we quantified 1,233 proteins per CSF sample (Fig 1B, Datasets EV1–EV3). The data acquired with DIA had 100% completeness for 385 proteins (26%), 75% for 1,050 proteins (71%), and 50% for 1,288 proteins (87%) (Fig 1C). The quantified protein intensities spanned over six orders of magnitude, in which the top ten most abundant proteins contributed 65% of total protein intensity of the entire 1,484 proteins in our dataset (Fig 1D). To

achieve such CSF proteome depth, extensive fractionation and depletion of abundant proteins often combined with isobaric labeling were previously required, with its associated disadvantages (preprint: Higginbotham *et al*, 2019; Sathe *et al*, 2019). For a single-shot CSF proteomics workflow that is amenable to high-throughput and large cohorts, this presents an unprecedented depth at high data completeness.

We investigated intra- and inter-assay variability of our automated CSF pipeline by repeated sample preparation (Materials and Methods), which revealed high reproducibility with over 1,000 proteins having inter-assay CVs below 20% (Fig EV2A and B, Datasets EV4 and EV5). This level of variability is much smaller than the proteome differences between subjects, as assessed by calculating the inter-individual variability within the cohorts. Here, only 225 proteins had a CV below 20% (Fig EV2C).

The availability of a large set of 197 CSF samples prompted us to investigate the relationship between different proteins in order to functionally interpret co-regulation of proteins that cluster with each other or with clinical parameters. The global protein correlation map (Wewer Albrechtsen *et al*, 2018) resulting from more than a million protein–protein comparisons highlighted eight main clusters of proteins which follow common functions or themes (Dataset EV6). For instance, neuronal annotation terms such as the gene ontology cellular compartments (GOCC) terms neuron projection, axon, and synapse were selectively enriched in the second largest cluster (Figs 1E and EV2D). Identification of neuronal proteins in the CSF highlights that proteins originating in the central nervous system accumulate in the CSF, thus making the CSF reflective of physiological or pathological proteome alteration in this organ.

Another cluster was enriched in blood plasma proteins relating to humoral immunity, the complement system or coagulation. Vascular proteins have been reported to be increased in AD brains while decreased in AD CSF (preprint: Higginbotham *et al*, 2019). However, apart from disease-associated effects such as a modulation of the blood–brain barrier, apparent alterations of blood protein abundances in CSF may be caused by blood contamination during CSF sampling which is hard to avoid entirely. Proteins are likely blood contaminants in CSF if they exhibit the same abundance profile across samples as known blood proteins and occur in the same abundance ratio to these blood proteins in CSF as in blood. Conversely, if a protein also found in blood does not correlate with the blood proteins, it may still be a genuine biomarker for AD. The global correlation map presents an efficient approach to distinguish biomarkers from contaminants (Geyer *et al*, 2019). Here, CSF signatures of proteins biologically relevant to AD clearly separated from protein clusters that are at higher risk to be contamination-associated (Fig 1E).

### Proteomics detects differences in CSF t-tau in individuals with or without AD and neuronal and widespread novel proteome alterations

In the Sweden and the Magdeburg/Kiel cohorts, AD was associated with drastic CSF proteome alterations, with 540 and 453 proteins significantly ($P < 0.05$) differing by AD status, respectively. These changes encompassed up- and down-regulated proteins, and significant proteins had a median absolute fold change of about 1.3-fold in both studies. The extensive brain atrophy apparent upon autopsy and the widespread brain proteome alterations harmonize well with the observed substantial alterations in the CSF proteome in AD and other neurodegenerative diseases (Hosp *et al*, 2017; preprint: Higginbotham *et al*, 2019). In all three cohorts, tau (gene name MAPT) was the most significantly or among the most significantly altered proteins between individuals with or without AD, with higher levels in AD (Fig 2A–C, Appendix Fig S1A–E). The fact that tau levels are elevated in AD CSF has been known for more than two decades but this important protein is not easily quantified in large proteomics discovery cohorts. Typically, tau quantitation by mass spectrometry has required extensive fractionation and depletion of abundant proteins, limiting throughput (preprint: Higginbotham *et al*, 2019; Sathe *et al*, 2019). Alternatively, targeting instead of discovery strategies can in principle quantify proteins such as tau in larger sample numbers (Barthélemy *et al*, 2016).

Cerebrospinal fluid is expected to reflect pathological alterations in functional classes of proteins. AD is characterized by synaptic dysfunction and neuronal cell death. Proteins associated with the gene ontology (GO) term "neuron projection" were indeed enriched in AD CSF compared with non-AD CSF ($P < 0.01$ in the all three cohorts; Fig 2A–C, Appendix Fig S1F). Likewise, proteins of the GO term "synapse part" were significantly enriched in AD CSF in the Sweden and Berlin cohorts ($P < 0.01$).

In the Berlin cohort, proteome alterations between AD and non-AD CSF were smaller with only 168 proteins exhibiting significantly ($P < 0.05$) different abundances (Figs 2C and 3A, Appendix Fig S1D and E). This finding concurs with the reduced biochemical separation of the AD and non-AD groups in the Berlin cohort based on clinical AD CSF biomarkers (Fig EV1B–K).

Despite fewer significantly different proteins, the Berlin cohort exhibited the same key features of the two other cohorts such as tau being a top outlier and the enrichment of neuronal and synaptic proteins. The second dominant outlier 14-3-3γ (gene name YWHAG) in the Berlin cohort was likewise enriched in AD CSF in the other cohorts. The family of 14-3-3 proteins is very abundant in the brain and has been implicated in neurodegenerative diseases, and increased levels of 14-3-3γ have been reported in AD brain tissue and CSF (Fountoulakis *et al*, 1999; Foote & Zhou, 2012; Sathe *et al*, 2019). Together, this shows a reduced but equivalent AD-associated effect on the CSF proteome in the Berlin cohort.

### Replication of AD-associated proteins across cohorts

As it had previously been challenging to establish biomarker panels that could be replicated across cohorts, we next assessed the consistency of AD-associated protein changes in this multi-cohort study. Of the significantly changed proteins described above, large proportions were consistent in their AD/non-AD association (Fig 3A and B, Dataset EV4). Comparing the Sweden and Magdeburg/Kiel cohorts, 89% (172/194 proteins) and 95% (102/107) were consistent at significance levels of $P < 0.05$ and $q < 0.05$, respectively. Likewise, comparing the Sweden and Berlin cohorts 95% (70/74) and 100% (16/16) were consistent applying the same criteria, respectively, equivalent to 93% (64/69) and 100% (14/14) comparing the Magdeburg/Kiel and Berlin cohort.

Furthermore, quantitative alterations of protein levels between AD and non-AD CSF were very consistent across the cohorts. AD/non-AD fold changes of proteins were highly correlated with Pearson's correlation coefficients at $r = 0.91$, $r = 0.80$, and $r = 0.90$ for the comparisons of Sweden and Magdeburg/Kiel, Sweden and Berlin, and Magdeburg/Kiel and Berlin, respectively (Fig 3C–E).

We assessed whether AD and non-AD samples clustered together independent of the cohort, based on either the global unfiltered CSF proteome profile, the less stringent ($P < 0.05$) intersection, or the more stringent ($q < 0.05$) intersection set of proteins significant in all three cohorts. After Z-scoring protein intensities within cohorts, unsupervised clustering clearly separated AD from non-AD groups in all three cases (global proteome, both intersection sets; Fig 3F and G, Appendix Fig S2A and B). In the $P < 0.05$ intersection set, 40 out of 43 proteins (93%) differed consistently in abundance by AD status, 35 of which had an elevated abundance in AD CSF and five an elevated abundance in non-AD CSF (Fig 3F, Appendix Fig S3A and B). We discuss these

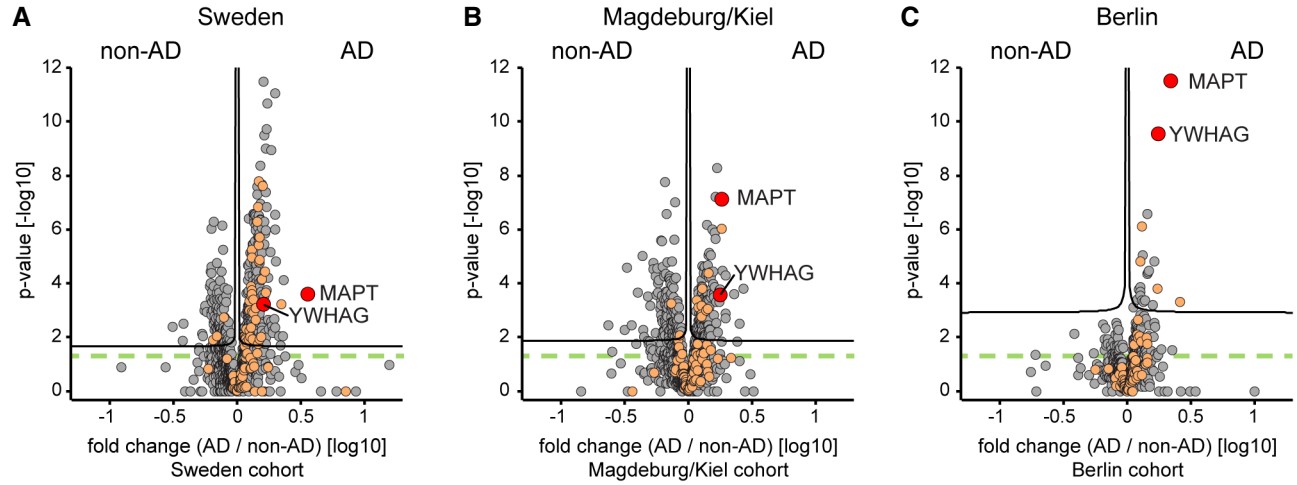

**Figure 2.  Differences in AD vs. non-AD CSF proteome in the three cohorts.**

A–C   Protein AD/non-AD fold changes plotted vs. statistical significance for Sweden (A), Magdeburg/Kiel (B), and Berlin (C) cohorts. Proteins associated with the GO annotation neuron projection labeled in orange. Proteins above the dashed green line are statistically significant (*P* < 0.05), and those above the black curves have a *q*-value below 0.05 (see Materials and Methods).

proteins as "the 40-protein signature" of AD in the remainder of this paper.

Next, we investigated if our results depended on the control groups in the Magdeburg/Kiel cohort and the Berlin cohort. The former controls were collected in Magdeburg or in Kiel, and in Berlin, the controls comprised subjective cognitive impairment patients and depression patients. Furthermore, the Kiel controls were younger than other cases or controls, and accordingly, their proteomes separated from the other non-AD controls (Fig EV1A, Appendix Fig S2A and B). Despite these differences, AD vs. non-AD fold changes of our 40-protein signature were independent of the specific non-AD control group subtype in these cohorts (Fig EV3A and B). To specifically investigate the effect of age and sex on the AD regulation of the 40-protein signature, we employed a linear regression model. After correction for age and sex in this way, the CSF abundance of all 40 proteins still significantly depends on AD status (Fig EV3C, Dataset EV7). Interestingly, CSF proteome alterations were of smaller magnitude in males compared to females in this study population.

Taken together, the "rectangular strategy" was able to discern AD-related alterations that reflect a small subset of the CSF proteome (< 50 proteins) from other cohort-specific effects comprising larger parts of the quantified CSF proteome (> 1,000 proteins) even in cohorts partially constrained by other biases such as age differences.

### AD-associated proteins in CSF are linked to neurodegeneration

Many proteins among our 40-protein signature have known or suspected links to AD or other neurodegenerative diseases (Fig 3F). For instance, PARK7 (protein/nucleic acid deglycase DJ-1) and SOD1 (superoxide dismutase 1) are risk genes for Parkinson's disease and amyotrophic lateral sclerosis, respectively (Bonifati *et al*, 2003; Renton *et al*, 2014). Notably, the two cellular superoxide dismutases SOD1 and SOD2 were more abundant in AD CSF

than in non-AD CSF, whereas the extracellular SOD3 was more abundant in non-AD CSF. Moreover, a genetic interaction of YWHAZ (14-3-3 protein ζ/δ) and BChE (butyryl cholinesterase) modulates the risk for AD (Mateo *et al*, 2008). CHI3L1 (protein YKL-40/chitinase-3-like protein 1), an astrocyte-derived protein, is elevated in AD CSF and discussed as a marker for progression from mild cognitive impairment to AD (Olsson *et al*, 2016; Baldacci *et al*, 2017). Similarly, fatty acid-binding protein 3 (FABP3) is elevated in AD CSF in our data and has been discussed as an AD CSF biomarker before (Sepe *et al*, 2018). CRYM (Ketimine-reductase mu-crystallin) has been reported as a modulator of huntingtin toxicity to striatal neurons in Huntington's disease (Francelle *et al*, 2015).

### Proteins differing by AD status correlate with CSF t-tau abundance and MMSE score

As CSF composition reflects brain health, proteins in CSF may differ between AD and control subjects and additionally correlate with severity of AD pathology as reflected by classical clinical parameters such as t-tau abundance in CSF. Indeed, in the total dataset of 1,484 proteins, 124 proteins correlated significantly (*P* < 0.05) with t-tau concentration, 19 of which had a correlation *q*-value below 0.05 (Fig 4A–D, Appendix Fig S4A, Dataset EV4). All 124 proteins showed a consistent directionality of positive or negative correlation across the three cohorts. The abundance of tau as measured by MS correlated well with the ELISA measurements (Pearson *r* = 0.82 for Sweden, *r* = 0.66 for Magdeburg, *r* = 0.68 for Berlin).

We next asked how our 40-protein signature correlated with clinical t-tau measurements. Indeed, a large fraction—29 of 40 proteins—significantly correlated with t-tau in each of the three cohorts, and the directionality of change was also as expected for the non-significant proteins (Fig 4A–E, Appendix Fig S4A). This is a substantial enrichment over the numbers expected by chance in this dataset (*P* < 0.0001, odds ratios 37). Upon adjustment for age, sex, and cohort in a linear regression model comprising all three cohorts,

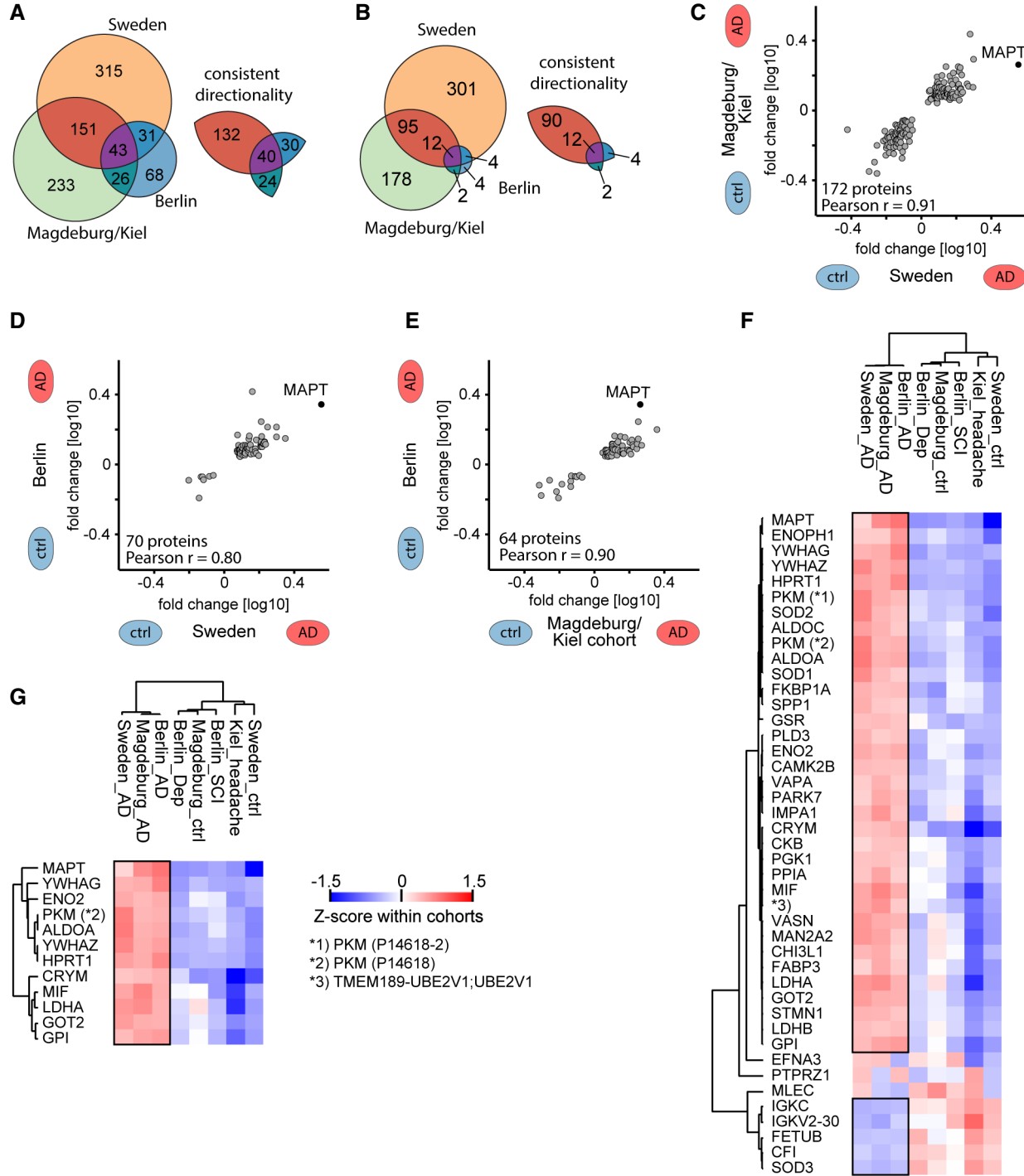

**Figure 3. CSF proteome alterations across the three cohorts.**

A, B  On the left, number of proteins that differ significantly (*P*-value < 0.05 in A; *q*-value < 0.05 in B) in abundance by AD status within each cohort. On the right, number of proteins thereof that have a consistent directionality of either elevated or reduced abundance in AD CSF in pairwise comparisons of cohorts.

C–E  Correlation of protein AD/non-AD fold changes in pairwise combinations of two cohorts each. Combinations are Sweden vs. Magdeburg/Kiel (C), Sweden vs. Berlin (D), and Magdeburg/Kiel vs. Berlin (E). Proteins included differ significantly (*P* < 0.05) and consistently in abundance by AD status in both cohorts each.

F, G  Proteins that differ significantly (*P* < 0.05 in E; *q* < 0.05 in F) in abundance by AD status across all three cohorts. Z-scored abundances of proteins in the AD and non-AD groups of all cohorts shown by the heat map (see Materials and Methods). Hierarchical clustering separates AD from non-AD groups. Pyruvate kinase PKM (PKM) was quantified in two isoforms, and UniProt IDs are given in parentheses. Black frames highlight proteins with consistent AD/non-AD fold changes across cohorts.

all 40 proteins were significantly associated with t-tau (Materials and Methods; Appendix Fig S4B, Dataset EV7).

Some of these proteins, including fructose-bisphosphate aldolase A (ALDOA), superoxide dismutase 1 (SOD1), and YKL-40/chitinase-3-like protein 1 (CHI3L1), have previously been reported to correlate positively with CSF t-tau levels (Dayon *et al*, 2018).

In the clinic, AD is routinely diagnosed by biochemical parameters or by cognitive tests. We therefore investigated the relation

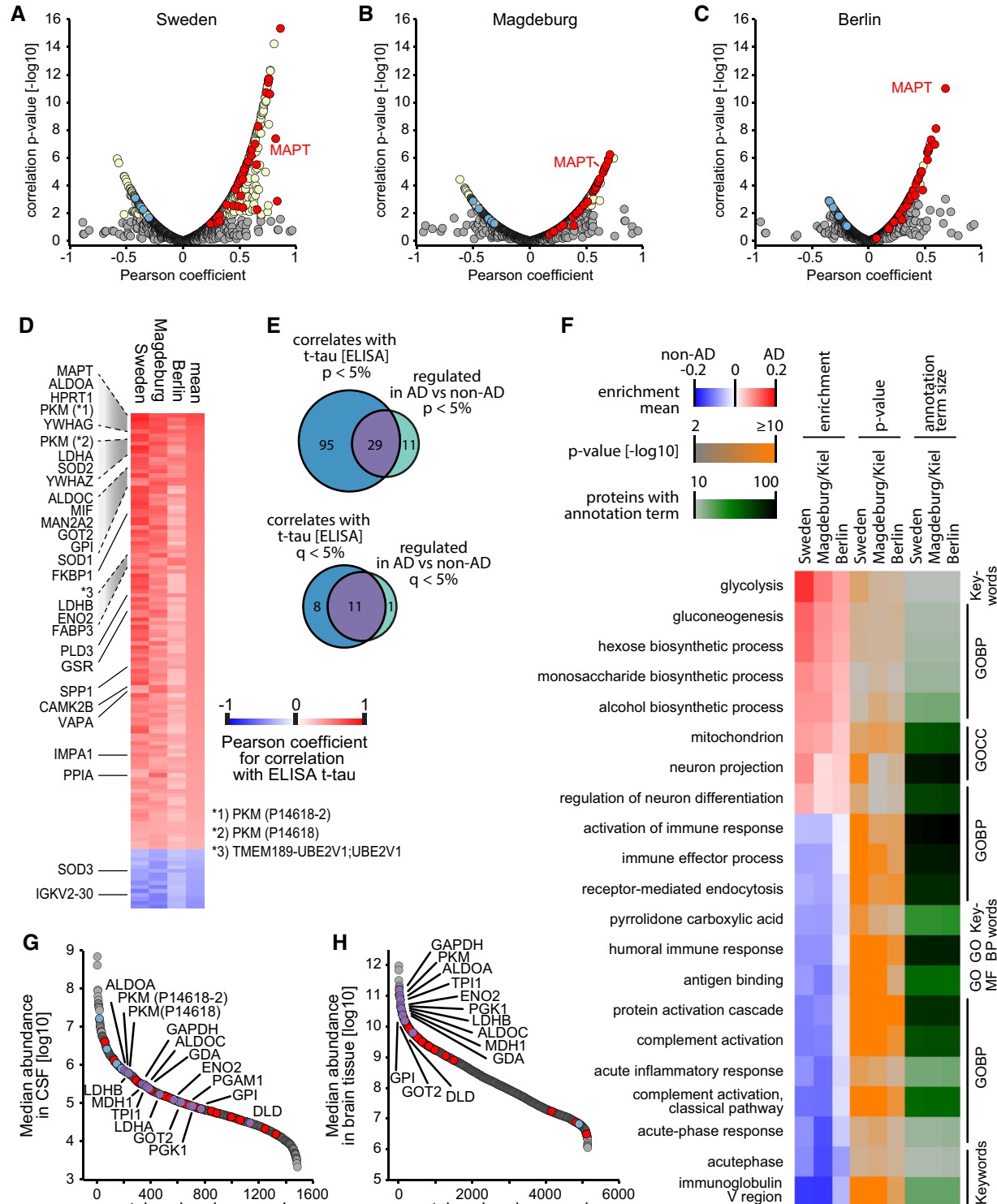

**Figure 4.**

**Figure 4. Protein correlation with t-tau measurements and analysis of annotation term enrichment.**

A–C   Correlation of proteins with ELISA-measured t-tau concentration across samples within the Sweden (A), Magdeburg (B), and Berlin (C) cohorts. Proteins with a *q*-value below 0.05 are labeled in yellow. Proteins of the 40-protein signature are colored in red for those with higher abundance in AD CSF and in blue for those with higher abundance in non-AD CSF.

D   Three-cohort summary of proteins significantly correlating with ELISA-measured t-tau. Protein names given for the 29 proteins out of the 40-protein signature with significant (*P* < 0.05) correlation in each of the three cohorts. Pyruvate kinase PKM (PKM) was quantified in two isoforms, and UniProt IDs are given in parentheses.

E   Overlap of proteins significantly differing by AD status with proteins significantly correlating to ELISA-measured t-tau.

F   Annotation enrichment in the AD versus non-AD fold change dimension. Terms with positive enrichment means are enriched in AD CSF over non-AD CSF. Conversely, terms with enrichment means below zero are enriched in non-AD compared with AD CSF. Annotations filtered for significance of enrichment (*P* < 0.05) and term size (10–100 proteins per term) in all three cohorts.

G, H   Protein abundance distribution of CSF (G) and brain (H) showing the abundances of AD-modulated CSF proteins. Proteins of our 40-protein signature are highlighted in red (elevated abundance in AD) and blue (elevated abundance in non-AD). Proteins linked to glucose metabolism are highlighted in purple and labeled.

between our proteomics results and the mini-mental state examination (MMSE) scores as a measure of cognitive performance, which were assessed in the Berlin cohort (Fig EV1L). In the literature, reference population means of MMSE scores were 29, 27, and 20 for cognitively normal, mild cognitive impairment (MCI), and AD participants, respectively (Chapman *et al*, 2016), while the MMSE scores in the Berlin cohort were 27.7 ± 1.9 (mean ± SD) for non-AD and 22.7 ± 4.5 for AD. Tau (MAPT), osteopontin (SPP1), and 14-3-3γ (YWHAG) were the top three proteins inversely correlating with the MMSE score (Fig EV4A). Osteopontin has already been reported to inversely correlate with the MMSE score in AD (Comi *et al*, 2010). Moreover, in our 40-protein signature proteins with higher abundance in AD CSF correlated negatively with the MMSE score and vice versa.

When stratifying the Berlin cohort into "high MMSE score" and "low MMSE score" groups over a cutoff range from 29 to 21, we obtained the greatest separation at a cutoff of 25. Reassuringly, MAPT and YWHAG were the top outliers and our 40-protein signature showed the expected association with the MMSE groups at all cutoff values in spite of the limited diagnostic performance of the MMSE evaluation (Fig EV4B–F) (Perneczky *et al*, 2006; Mitchell, 2009; Arevalo-Rodriguez *et al*, 2015). Thus, CSF protein signatures linked to biochemically defined AD also associate with cognitive performance.

## Neuronal and glycolytic signature in AD CSF

To identify biological signatures in the AD-associated proteome alterations, we performed an annotation enrichment analysis of functional terms (GO biological process, GO cellular compartment, UniProt Keywords) in the global proteome AD/non-AD fold changes. We obtained 21 annotation terms below a *P*-value of 0.05, all of which showed consistency across the three cohorts (Materials and Methods, Fig 4F). Terms including "neuron projection" and "regulation of neuron differentiation" underline the neuronal signature in the AD CSF proteome. Interestingly, glycolysis and gluconeogenesis presented as top terms with enrichment in AD CSF in this unbiased analysis. This concurs with the presence of glycolytic proteins in our 40-protein signature. These include fructose-bisphosphate aldolase A (ALDOA) and C (ALDOC), pyruvate kinase PKM (PKM), γ-enolase (ENO2), aspartate aminotransferase, mitochondrial (GOT2), phosphoglycerate kinase 1 (PGK1), L-lactate dehydrogenase A chain (LDHA), and B chain (LDHB) (Fig 3F). Moreover, other glycolytic

proteins in the dataset not passing the significance cutoffs nevertheless uniformly followed the same trend of elevated abundances in AD CSF (Appendix Fig S5). Glycolytic proteins may originate from astrocytes as glycolysis in the brain is mainly performed by these cells to provide lactate for oxidative phosphorylation in neurons (Bélanger *et al*, 2011; preprint: Higginbotham *et al*, 2019). Furthermore, the GO cellular compartment annotation term "mitochondrion" was also enriched in AD CSF, and mitochondrial dysfunction is a known hallmark of AD (Querfurth & LaFerla, 2010). When we mapped the up-regulated proteins of our 40-protein signature onto a deep human brain proteome (Carlyle *et al*, 2017), their corresponding abundance in brain was generally in the more abundant range (Fig 4G and H). This observation is consistent with mechanisms in which cellular proteins are released into the CSF by tissue damage-associated loss of membrane integrity, exosome release, or others.

## Further confirmation of AD-associated proteome alterations in an independent cohort

After completion of our study, a related preprint appeared (preprint: Higginbotham *et al*, 2019). Similarly to our study, the authors investigated proteomic profiles in a study of 20 AD cases and 20 controls, although they used a different experimental workflow. CSF samples were depleted, digested, chemically labeled for multiplexing by an isobaric tag, fractionated, and analyzed by mass spectrometry, achieving a remarkable depth of quantitation. A second cohort, consisting of 33 AD and 32 controls and 30 asymptomatic cases, was also measured, although with a somewhat different method and a reduced proteome depth. Many AD-associated CSF signatures observed in our study including the glycolytic signature, the neuronal signature, and the 14-3-3 protein signature are also reported in the manuscript. This provides additional evidence for these signatures to be AD-associated from independent cohorts identified by a different experimental approach.

To determine a panel of consistently AD-regulated proteins and to assess inter-study consistency in more detail, we downloaded the available data and compared them to our data. As tau was not contained in the second cohort dataset and only 31 proteins significantly differed by AD status in both cohorts of that independent study, we limited our comparison to the 20 AD cases versus 20 controls cohort by Higginbotham *et al*. This dataset contained 2,875 proteins quantified in at least half of the samples and 528 proteins thereof differed significantly (*P* < 0.05) by AD status. Notably,

despite the different proteome depth the number of proteins that differed by AD status is similar to the proteins that differed significantly by AD status in the Sweden (540) and Magdeburg/Kiel (453) cohorts. These similar numbers in three out of four cohorts suggest that both proteomic approaches cover a substantial part of the CSF proteome signature related to AD.

Out of our 40-protein signature, 38 proteins were contained in the dataset of this independent study and 26 of 38 (68%) thereof were also significant (Fig EV5A, Dataset EV4). This is a highly significant enrichment among all significant proteins in the dataset of that independent study (odds ratio 10, $P < 0.0001$, Fig EV5B). The directionality of abundance elevation in either AD or non-AD CSF was consistent across studies for these 26 core proteins (Fig EV5C). Moreover, quantitative fold change agreement was high (Pearson $r = 0.76$; Fig EV5D). Among these 26 proteins, only one protein, fetuin-B (FETUB), had an elevated abundance in non-AD CSF, while 25 proteins were elevated in AD CSF including tau, glycolysis-related proteins, 14-3-3 proteins, protein/nucleic acid deglycase DJ-1 (PARK7), superoxide dismutase 1 (SOD1), fatty acid-binding protein 3 (FABP3) and hypoxanthine-guanine phosphoribosyltransferase (HPRT1). Taken together, AD-associated protein signatures identified in our work are validated in a completely separate study using an independent cohort and different experimental strategy.

### AD classification by machine learning on the CSF signature

Next, we next assessed if the MS intensities of the set of 26 core proteins which overlap between our and the Higginbotham studies could be applied to classify participants by AD status using machine learning and we explored a variety of machine learning models. First, to determine feature importance, we employed a decision tree and found that a model with a maximum depth of six levels, using the intensities of 14 proteins could correctly classify the participants in the three studies by AD status. A visualization of the decision tree revealed that levels of tau itself were at the root, followed by the glycolytic enzyme pyruvate kinase PKM (PKM), and macrophage migration inhibitory factor (MIF) at the next level (Fig 5A). As protein intensities are correlated, a decision tree could potentially rank proteins differently depending on its initial state. However, when repeatedly training the decision tree ($n = 10,000$) with random initial states and also shuffling the dataset, the root of the tree remained similar (MAPT at rank 1 in all cases, PKM at rank 2 or 3 in 82.8%, and MIF at rank 2 or 3 in 84.3% of all cases, respectively). This underlines the importance of these three proteins among the CSF proteome as indicators of AD.

To test models for generalizability, we considered several tree-based ensemble methods. We trained six commonly used methods (AdaBoost, Bagging, ExtraTrees, GradientBoosting, RandomForest, and XGBoost) on the intensities of the 14 proteins selected by the decision tree above such that the tree needed to completely classify the participants. The protein intensities were randomly shuffled and split using a k-folds cross-validator ($k = 6$) into six training/test splits. Accordingly, shuffling entailed mixing of patients from different cohorts but each sample was in the testing dataset exactly once. For each method, we performed cross-validation and determined a receiver operating characteristic (ROC) curve.

All classifiers reached an area under the ROC curve (AUC) of at least 0.84. XGBoost had the best performance with a mean AUC of 0.91 and was selected for further analysis. To determine the optimal number of features, we iteratively added them in them in their order of importance in the decision tree. The overall model performance increased with the number of proteins and reached a plateau at six proteins (MAPT, PKM [P14618-2 isoform], MIF, IMPA1, YWHAZ, and ALDOC), which we selected for the final model.

To assess the performance of our final model as a predictive test we again used k-fold cross-validation in six different training/test splits. The different splits exhibited good agreement with each other at AUC's ranging from 0.87 to 0.98, indicating robustness of classification (Fig 5B). We then determined the overall confusion matrix combining the six splits ("net reclassification", the number of correctly and incorrectly classified participants) (Fig 5C). In total, 72 out of 88 AD patients and 95 out of 109 non-AD patients were correctly identified, corresponding to a sensitivity of 82% and a specificity of 87%.

## Discussion

We have combined advanced sample preparation, cutting-edge mass spectrometry hardware, acquisition schemes, MS data processing and bioinformatic analysis and optimized it for CSF to build a high-performance CSF proteomics workflow amenable to high-throughput and large cohorts. About 1,500 proteins can be quantified and over 1,000 with intra- and inter-assay coefficients of variation (CVs) below 20%. Using this technology, we identified known biomarkers such as tau as top candidates as well as a range of novel potential biomarkers. Harnessing this pipeline, we compared AD and non-AD CSF in three independent cohorts. This led to a 40-protein signature whose members are consistently up- or down-regulated in AD CSF vs. non-AD CSF across the three cohorts.

Cases and controls in two of our cohorts separated better on the basis of clinical AD CSF biomarker concentrations (t-tau, p-tau$_{181}$, A$\beta_{1-42}$, A$\beta_{1-40}$) than in the third one. Likewise, AD-associated differences in the CSF proteome were smaller and fewer protein alterations were statistically significant in that third cohort. The attenuated separation according to clinical CSF values suggests that this third cohort comprised milder AD cases and early-stage AD patients in the non-AD group just below the cutoff values. This would lead to the attenuated overall differences in the CSF proteome profile between the AD and the non-AD groups that we observe.

There is no universally accepted AD classification system; however, various different integration schemes of clinical AD CSF biomarkers have been explored (Bloudek *et al*, 2011; Ferreira *et al*, 2014; Ritchie *et al*, 2017). Using the Hulstaert index, a variation of the A$\beta_{1-42}$/t-tau ratio, for AD classification of the three cohorts we obtained largely the same, but fewer statistically significant potential marker proteins compared to our uniform AD classification (Appendix Fig S6A–D, Materials and Methods) (Hulstaert *et al*, 1999; Molinuevo *et al*, 2013; Vos *et al*, 2013). Furthermore, the mini-mental state examination (MMSE) cognitive test was performed in one of our cohorts. It was encouraging to find the proteomic outliers identified by analysis of biochemically defined AD CSF to be associated with the MMSE score performance.

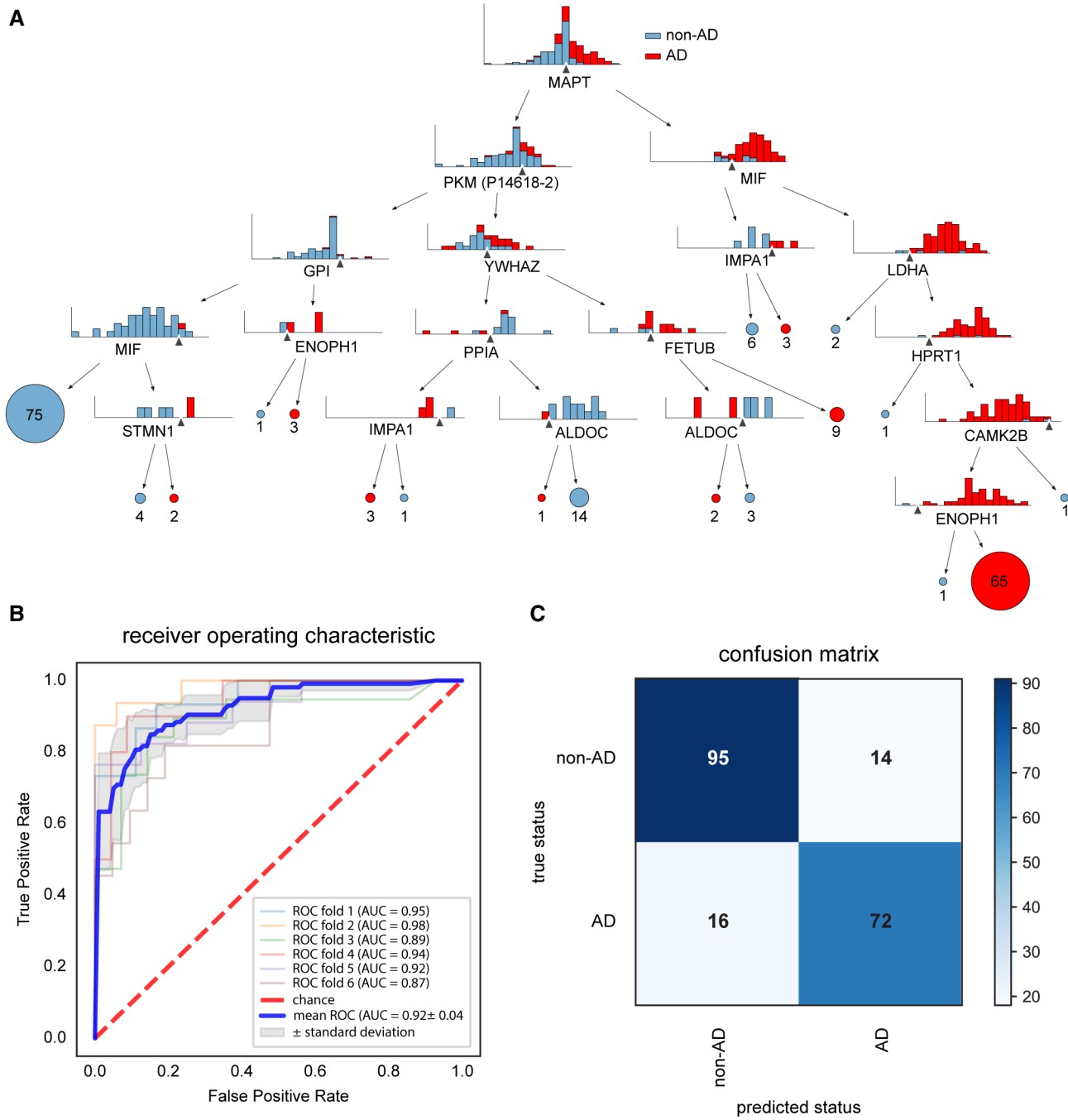

**Figure 5. Machine learning separates AD from non-AD CSF at high performance.**

A  Decision tree to classify AD vs. non-AD participants based on the protein levels of a core 26 protein set. Splits are indicated by black triangles. A tree with a minimum depth of six can correctly classify the participants by AD status.

B  Receiver operating characteristic (ROC) curve for the model based on XGBoost. The diagonal line indicates random performance. Blue line represents the mean performance of the model when trained on six stratified train—test splits (k-fold). The gray areas represent the standard deviation of ROC values.

C  Confusion matrix indicating model performance when predicted on the test split of the cross-validation. Overall accuracy is 0.85.

Another general challenge in biomarker discovery studies are cohort-specific effects. This relates particularly to multi-centric studies with distinct inclusion criteria for cases and controls. Despite cohort-specific effects and attenuated AD/non-AD differences in the third cohort of our study, proteins that statistically significantly differed by AD status in multiple cohorts exhibited very good qualitative and quantitative cross-cohort agreement in their AD modulation. A signature of 40 CSF proteins was consistently associated with AD status and showed high correlation values of protein fold changes across cohorts. When further combined with a recent,

independent effort on bioRxiv (preprint: Higginbotham *et al*, 2019; Johnson *et al*, 2020), which used different MS technology, this resulted in a set of 26 core proteins consistent across four independent cohorts. This highlights the power of the "rectangular strategy" study design in discerning cohort-specific from pathological effects for biomarker discovery.

Our relatively large dataset with nearly 200 participants prompted us to explore machine learning for the purpose of assessing AD status on the basis of the levels of the 26 core proteins. We found that an ensemble method-based classifier reached high specificity (87%) and sensitivity (82%), while showing promising generalizability. Intriguingly, tau itself, one of the glycolysis-related proteins, and an immunological factor were selected by the machine learning algorithm as the most important features for classification, proving further validation of our biomarker panel and biomarker identification pipeline. The modeling also indicated that additional and more uniform training data could further improve diagnostic performance. Furthermore, additional clinical data, such as cognitive assessments, can naturally be incorporated in this framework.

In the list of the 26 core proteins, several have known links to neurodegeneration such as protein/nucleic acid deglycase DJ-1 (PARK7) and superoxide dismutase 1 (SOD1) or genetic interaction links to AD like 14-3-3 protein $\zeta/\delta$ (YWHAZ) (Bonifati *et al*, 2003; Mateo *et al*, 2008; Renton *et al*, 2014). Likewise, the set also contains the tentative AD biomarker CHI3L1 (protein YKL-40) likely reflecting astrocytic activation (Olsson *et al*, 2016; Baldacci *et al*, 2017). Moreover, we identify a number of glucose metabolism-associated proteins elevated in AD CSF in line with other reports (Dayon *et al*, 2018; preprint: Higginbotham *et al*, 2019; Sathe *et al*, 2019). These glycolytic proteins and other AD-associated proteins in CSF are highly abundant in brain and could be released into CSF from brain tissue. Regardless of the mechanism of accumulation in the CSF, the utility of abundant cellular proteins as markers is generally accepted in clinical practice. In the plasma proteome, this is demonstrated by troponin levels indicative of acute myocardial infarction (Keshishian *et al*, 2015) and liver proteins indicative of fatty liver disease (Niu *et al*, 2019).

The fact that CSF proteomics is now able to detect brain-derived proteins and determine protein signatures consistent across multiple independent multi-centric cohorts sets the stage for future biomarker discovery studies in neurodegenerative diseases. Next steps should include investigating the added diagnostic value of the AD CSF protein signature when combined with established diagnostic criteria in the clinic, preferably in a machine learning framework. Further, we speculate that the workflow presented here would be highly suited for the discovery of additional clinically and etiologically relevant biomarkers. There is a great need for early diagnosis, prognosis, and treatment efficacy biomarkers (Winblad *et al*, 2016). Further studies are warranted assessing the relevance of these proteins in prospective studies of dementia-free individuals in midlife with repeated brain imaging, cognitive testing, and long-term follow-up for dementia incidence. Recent developments in MS-based proteomics now enable fast and efficient quantitative readout of relatively large panels of proteins in a targeted or "globally targeted" manner (Abbatiello *et al*, 2013; Wichmann *et al*, 2019). This may enable the use of MS-based proteomics not only for the discovery of disease-associated protein patterns but also for routine clinical tests (Geyer *et al*, 2017).

# Materials and Methods

## Study populations

Three cohorts of AD and non-AD control CSF samples were obtained, one from Sweden, one originating from the German cities of Magdeburg and Kiel (through the Harvard T. H. Chan School of Public Health), and one from Berlin. The CSF concentration values of the clinical AD biomarkers t-tau, p-tau$_{181}$, A$\beta_{1–42}$, and A$\beta_{1–40}$ were available as follows: t-tau, p-tau$_{181}$, A$\beta_{1–42}$ for the Sweden cohort; t-tau, p-tau$_{181}$, A$\beta_{1–42}$, and A$\beta_{1–40}$ for the Magdeburg cohort; and t-tau, A$\beta_{1–42}$, and A$\beta_{1–40}$ for the Berlin cohort.

Sweden CSF samples were obtained from patients with cognitive impairment at several memory clinics in western Sweden. De-identified diagnostic remnant CSF material was used in this study, which was approved by the Gothenburg ethics committee. The AD and non-AD groups as classified by the primary AD criteria of this study were well separated biochemically based on the clinical AD CSF biomarkers. CSF biomarker levels were measured using the INNOTEST assays (Fujirebio, Ghent, Belgium) in the Clinical Neurochemistry Laboratory, Sahlgrenska University Hospital, Mölndal, Sweden, by board-certified laboratory technicians who were blinded to clinical data. The laboratory procedures were accredited by the Swedish Board for Accreditation and Conformity Assessment (SWEDAC).

Magdeburg CSF samples originated from patients at the outpatient memory clinic at the Otto-von-Guericke University Magdeburg. CSF biomarker levels were measured at the site of collection using commercially available INNOTEST ELISA kits (Fujirebio, Ghent, Belgium). The AD and non-AD groups as defined by our primary AD classification criteria were well separated biochemically based on the clinical AD CSF biomarkers. The local ethics committee approved the use of the CSF samples. Additional control samples from Kiel were acquired from patients treated at the emergency department at the University Hospital Schleswig-Holstein. Informed consent for scientific analysis of diagnostic remnant samples collected for clinical care and ethics committee approval for use of the samples were obtained.

Berlin CSF samples were obtained from patients at the Memory Clinic of Charité Universitätsmedizin Berlin. The clinical AD biomarkers t-tau, A$\beta_{1–42}$, and A$\beta_{1–40}$ were measured at the site of collection. The V-PLEX A$\beta$ Peptide Panel 1 (6E10) Kit (Meso Scale Diagnostics, Rockville, MD, USA) was used for A$\beta$ peptide quantitation and the INNOTEST hTAU Ag (Fujirebio Germany GmbH, Hannover, Germany) for tau. The AD and non-AD groups as defined by our primary AD classification criteria were moderately separated biochemically based on the clinical AD CSF biomarkers. CSF collection was standardized as described elsewhere (Schipke *et al*, 2011). The local ethics committee approved the use of the CSF samples. All participants provided written informed consent.

## Primary AD classification

To enable uniform analysis, we standardized classification of AD and non-AD for the different cohorts uniformly based on the CSF concentrations of t-tau, A$\beta_{1–42}$, and A$\beta_{1–40}$ for the Sweden, Magdeburg, and Berlin cohorts. Patients were classified as AD if the t-tau concentration was above 400 ng/l and the A$\beta_{1–42}$ concentration

below 550 ng/l or the $A\beta_{1-42}/A\beta_{1-40}$ ratio was below 0.065. The t-tau criterion and at least one of the two $A\beta$ criteria had to be met for a patient to be classified as having AD and patients were classified as not having AD otherwise. The classification here is derived from a classification according to the cutoffs of t-tau being higher than 400 ng/l, p-tau$_{181}$ higher than 60 ng/l, and $A\beta_{1-42}$ lower than 550 ng/l (Sjogren *et al*, 2001; Hansson *et al*, 2006). We additionally included the CSF $A\beta_{1-42}/A\beta_{1-40}$ ratio as it has a superior diagnostic performance than the $A\beta_{1-42}$ concentration alone (Spies *et al*, 2010; Dubois *et al*, 2014; Niemantsverdriet *et al*, 2017). Participants with missing information on the CSF t-tau or $A\beta_{1-42}$ concentration in the Sweden, Magdeburg, or Berlin cohort were excluded. Kiel CSF samples originated from young patients (32.0 ± 17.1 years, median ± SD) treated at an emergency department with no indications of AD or other neurodegenerative diseases. Thus, we included these samples as non-AD controls despite the missing clinical biomarker CSF concentrations.

## Hulstaert index

The Hulstaert index for AD classification is a variant of the $A\beta_{1-42}/$t-tau ratio with improved diagnostic performance (Molinuevo *et al*, 2013). It is calculated as $A\beta_{1-42}/(240 + (1.18{*}\text{t-tau}))$ using ng/l concentrations, and samples below a cutoff value of one are classified as AD (Hulstaert *et al*, 1999). We performed an independent analysis using the Hulstaert index instead of our uniform classification. As shown in Appendix Fig S6, the results overlap almost completely; however, the Hulstaert index, although less stringent in AD inclusion, leads to a smaller number of significantly different proteins.

## Clinical AD diagnosis

Information about clinical AD status, i.e. the diagnosis of symptomatic AD according to site-specific criteria, was available for the Magdeburg, Kiel, and Berlin cohorts. At these sites, clinical AD diagnoses had been reached by assessing the clinical presentation of patients according to distinct guidelines.

In Magdeburg, the clinical AD diagnosis was based on the patient's clinical presentation using the National Institute of Neurological and Communicative Disorders and Stroke—Alzheimer's Disease and Related Disorders Association (NINCDS-ADRDA) criteria (McKhann *et al*, 2011). The clinical evaluation included the CERAD (Consortium to Establish a Registry for Alzheimer's Disease) neuropsychological test battery and magnetic resonance imaging (Morris *et al*, 1989). AD and control subjects had no clinical signs of stroke, epilepsy, or other neurodegenerative diseases. For the clinical diagnosis of AD, local concentration cutoffs for core AD biomarkers were used; however, fulfillment of the cutoff criteria was considered indicative but not sufficient for an AD diagnosis which also depended on the patient's clinical presentation. AD was considered likely if the criteria p-tau$_{181}$ > 80 ng/l and t-tau > 450 ng/l were simultaneously met. Likewise, AD was considered likely if the criteria $A\beta_{1-42}$ < 485 ng/l and the amyloid ratio $A\beta_{1-42}/A\beta_{1-40}$ < 0.06. Non-AD control patients underwent CSF withdrawal to exclude neuroinflammation and dementia. Control subjects showed no signs of neurodegeneration and had normal CSF parameters regarding cell count, protein content, and lactate

concentration. All but one of 26 biochemically defined AD cases according to our primary AD classification study also had a clinical AD diagnosis, while none of the non-AD controls had a clinical AD diagnosis.

Kiel CSF samples originated from patients presenting with acute headache. No patient had an AD diagnosis or showed clinical indications of neurodegenerative diseases. CSF sampling was performed to exclude meningitis which is not present in any subject in this study. Subjects with a history of dementia, systemic or CSF inflammatory signs, and blood–brain barrier dysfunction (CSF-to-serum albumin ratios ≥ 9 × 10$^3$) were excluded, and clinical diagnoses were diverse and predominantly migraine, headache, common cold or sinusitis or skin sensation disturbance (Koch *et al*, 2017).

In Berlin, patients were diagnosed as having AD based on the clinical presentation according to the American Psychiatric Association guidelines, the Diagnostic and Statistical Manual of Mental Disorders (DSM), version DSM-5. Diagnoses were reached at a consensus panel composed of psychiatrists, neurobiologists, and neuropsychologists according to the DSM-5. Specifically, patients' relevant medical history, standard cognitive and functional measurements (e.g., MMSE), CSF biomarker values for t-tau and amyloid peptides, and cMRI findings were examined in parallel. For the clinical diagnosis of AD, site-specific CSF concentration cutoffs for core AD biomarkers were used. Under these conditions, the following CSF biomarker values were rated as indicative of AD: $A\beta_{1-42}$ < 600 ng/l or $A\beta_{1-42}/A\beta_{1-40}$ ratio ≤ 0.060 (in 2014 and before) or $A\beta_{1-42}/A\beta_{1-40}$ ratio ≤ 0.065 (from 2015 on), in addition to t-tau > 350 ng/l. Again, however, fulfillment of these cutoff criteria was considered indicative but not sufficient for an AD diagnosis which also depended on the patient's clinical presentation. Out of 33 biochemically defined AD cases according to of our primary AD classification, 24 also had a clinical AD diagnosis at the time of CSF withdrawal, while none of the non-AD controls had a clinical AD diagnosis. For three of the nine biochemically defined AD cases without a clinical AD diagnosis, the medical records included additional clinical information or information collected months to years after the CSF withdrawal as the patient returned to the clinic again. These three patients either developed clinical AD within 2 years, presented with mild cognitive deficiencies of the AD type or a "not yet specified neurodegenerative disease".

## Sample preparation

The sample preparation was optimized for CSF on the basis of our Plasma Proteome Profiling workflow (Geyer *et al*, 2016). CSF was aliquoted in 96-well plates and processed with an automated set-up on an Agilent Bravo liquid handling platform. In total, 40 μl of CSF was mixed with 40 μl PreOmics lysis buffer (PreOmics GmbH) for reduction of disulfide bridges, cysteine alkylation, and protein denaturation at 95°C for 10 min. After a 10-min cooling step, 0.2 μg trypsin and 0.2 μg LysC were added to each sample and digestion was performed at 37°C for 4 h. Peptides were purified on two 14-gauge StageTip plugs according to the PreOmics iST protocol (https://preomics.com/products). The StageTips were centrifuged using an in-house 3D-printed StageTip tray at 1,500 *g* for washing and elution. The eluate was completely dried using a SpeedVac centrifuge at 45°C (Eppendorf, Concentrator plus), resuspended in

10 µl buffer A* (2% v/v acetonitrile, 0.1% v/v trifluoroacetic acid, and stored at −20°C. Upon thawing, samples were shaken for 1 min at 2,000 rpm (thermomixer C, Eppendorf). Peptides were then subjected to LC-MS/MS analysis.

Additionally, for library generation for the DIA measurements, peptides of the Sweden cohort were pooled into one AD sample pool and one non-AD sample pool of 75 µg each. Peptide concentration was measured spectroscopically by absorbance at 280 nm (Nanodrop 2000, Thermo Scientific). The AD sample pool and the non-AD sample pool were fractionated into 24 fractions each by high-pH reversed-phase chromatography on the "spider fractionator" (Kulak *et al*, 2017). Fractions were completely dried and resuspended in 10 µl buffer A*. To determine coefficients of variation, five aliquots of a CSF pool on one plate were subjected to sample preparation (intra-plate) and this was repeated on three different days (inter-plate).

## Mass spectrometry analysis

Samples were measured using an EASY-nLC 1200 (Thermo Fisher Scientific) coupled to a Q Exactive HF-X Orbitrap mass spectrometer (Thermo Fisher Scientific) via a nano-electrospray ion source (Thermo Fisher Scientific). Purified peptides were separated on 50 cm UHPLC columns with an inner diameter of 75 µm packed in-house with ReproSil-Pur C18-AQ 1.9 µm resin (Dr. Maisch GmbH). In total, 500 ng of purified peptide in buffer A* was loaded onto the column in buffer A (0.1% v/v formic acid) and eluted at a flow rate of 300 nl/min and a temperature of 60°C by a linear 80-min gradient from 5% to 30% buffer B (0.1% v/v formic acid, 80% v/v acetonitrile), followed by a 4-min increase to 60% B, a further 4-min increase to 95% B, a 4-min plateau phase at 95% B, a 4-min decrease to 5% B, and a 4-min wash phase of 5% B. To acquire MS data, the data-independent acquisition (DIA) scan mode was used for single-shot patient samples, whereas the fractionated samples of the AD pool and non-AD pool were acquired with a top12 data-dependent acquisition (DDA) scan mode. Both acquisition schemes were combined with the same liquid chromatography gradient. The mass spectrometer was operated by the Xcalibur software (Thermo Fisher). DDA scan settings on full MS level included an ion target value of $3 \times 10^6$ charges in the 300–1,650 m/z range with a maximum injection time of 20 ms and a resolution of 60,000 at m/z 200. At the MS/MS level, the target value was $10^5$ charges with a maximum injection time of 60 ms and a resolution of 15,000 at m/z 200. For MS/MS events only, precursor ions with 2–5 charges that were not on the 20 s dynamic exclusion list were isolated in a 1.4 m/z window. Fragmentation was performed by higher-energy C-trap dissociation (HCD) with a normalized collision energy of 27 eV. DIA was performed with one full MS event followed by 33 MS/MS windows in one cycle resulting in a cycle time of 2.7 s. The full MS settings included an ion target value of $3 \times 10^6$ charges in the 300–1,650 m/z range with a maximum injection time of 60 ms and a resolution of 120,000 at m/z 200. DIA precursor windows ranged from 300.5 m/z (lower boundary of the first window) to 1649.5 m/z (upper boundary of the 33$^{rd}$ window). MS/MS settings included an ion target value of $3 \times 10^6$ charges for the precursor window with an Xcalibur-automated maximum injection time and a resolution of 30,000 at m/z 200.

## Mass spectrometry data processing

The MS data of the fractionated pools (DDA MS data, 24 AD fractions, 24 non-AD fractions) and the single-shot subject samples (DIA MS data, all samples from all three cohorts) were used to generate a DDA-library and direct-DIA-library, respectively, which were computationally merged into a hybrid library in the SpectroMine software, version 1.0.21621.8.15296 (Biognosys AG, Schlieren, Switzerland). The hybrid library contained 33,392 precursors linked to 23,855 unique peptides considering peptide modifications or 17,301 unique peptides based on the amino acid sequence corresponding to 2,733 protein groups. The hybrid spectral library was used to search the MS data of the single-shot patient samples in the Spectronaut software, version 12.0.20491.9.26669 (Biognosys AG), for final protein identification and quantitation. All searches were performed against the human UniProt reference proteome of canonical and isoform sequences with 93,786 entries downloaded in March 2018. Searches used carbamidomethylation as fixed modification and acetylation of the protein N-terminus, oxidation of methionines and deamidation of asparagine or glutamine as variable modifications. Default settings were used for other parameters. In brief, a trypsin/P proteolytic cleavage rule was used, permitting a maximum of two miscleavages and a peptide length of 7–52 amino acids. Protein intensities were normalized using the "Local Normalization" algorithm in Spectronaut based on a local regression model (Callister *et al*, 2006). Spectral library generation stipulated a minimum of three fragments per peptide, and maximally, the six best fragments were included. A protein and precursor FDR of 1% were used and protein quantities were reported in samples only if the protein passed the filter ("Q-value sparse" mode data filtering).

## Bioinformatics data analysis

Data analysis was mainly performed in the Perseus environment version 1.6.1.3 but also in version 1.6.0.9 for correlation analysis and version 1.5.2.11 for Venn diagram analysis (Tyanova *et al*, 2016). Proteins with < 20 observations across the entire dataset were excluded, reducing the dataset from 1,542 to 1,484 proteins. Protein intensities were log$_{10}$-transformed for further analysis, apart from correlation and coefficient of variation analysis. All t-tests performed were two-sided and unpaired. False discovery rate (FDR) control to account for multiple hypothesis testing in statistical tests was performed by a permutation-based model in conjunction with a SAM-statistic with an s0-parameter of 0.001 (Tusher *et al*, 2002). Annotation term enrichment was performed with the 1D enrichment tool in Perseus separately for each cohort (Cox & Mann, 2012). Annotation terms were filtered for terms with a *P*-value cutoff of 0.5% in each cohort. Moreover, terms comprising less than 10 or more than 100 proteins in our dataset of 1,484 proteins were excluded because we found that annotation enrichment analysis is often dominated by very small or large but not meaningful terms. Hierarchical clusters were generated using the built-in tool in Perseus. When protein abundances were reported on the group level (e.g. Sweden AD), Z-scoring across samples either within the cohort or across cohorts (for all 197 samples) was performed as stated in the figure legends and the median Z-score was taken as group abundance. Sample groups (e.g. Sweden AD) were clustered based on

Pearson's correlation coefficient, while proteins were clustered based on Euclidian distance unless ranked by the three-cohort mean.

A deep human brain proteome was used for comparison to the CSF proteome, and 753 proteins were matched based on ensemble identifiers (Carlyle et al, 2017, supplementary table 5). For generation of the abundance distribution curves, median protein abundances across all samples within a proteome were used. For the comparison of AD CSF proteomes with the independent report (preprint: Higginbotham et al, 2019), data for the CSF1 dataset were downloaded from bioRxiv.org. Proteins were matched to our data based on UniProt protein identifiers, apart from of MAPT, ALDOA, and SOD2 which were matched based on gene names. Fisher's exact test in combination with the Baptista–Pike method was used in GraphPad Prism version 7.03 to assess the significance of enrichment and odds ratios in contingency table settings. This included the analysis of association of t-tau concentration-correlated proteins with proteins differing by AD status and the analysis of enrichment of AD-regulated proteins identified in this study among the proteins differing significantly ($P < 0.05$) by AD status in the Higginbotham CSF1 dataset.

We used linear regression analysis computed in RStudio version 1.2.5033 using R version 3.6.3 and assessed the association of $\log_{10}$-transformed protein intensities first with AD status (Fig EV3C) and second with the $\log_{10}$-transformed ELISA-measured CSF t-tau concentration (Appendix Fig S4B), adjusting for age, sex, and cohort (Sweden, Magdeburg/Kiel, or Berlin) in both models. To compare estimators of binary (AD status, sex) to those of continuous variables (age, t-tau concentration [$\log_{10}$]), the estimators for continuous variables (i.e. per 1 year [age] and per 1 unit in $\log_{10}$ space of t-tau concentration/[ng/l]) were multiplied with the interquartile range (IQR) of the variable for plotting. IQRs for age were eleven years for the complete dataset (Fig EV3C) and 9 years for the reduced dataset excluding the Kiel samples due to missing t-tau concentration values (Appendix Fig S4B). The t-tau concentration 75% and 25% quantiles were 802 ng/l and 275 ng/l, respectively, corresponding to an interquartile range of 527 and 0.4648 in linear and $\log_{10}$ space, respectively (Appendix Fig S4B). Regression coefficients for age and sex were displayed in the heat map if the $P$-value for these estimators was below 0.05. All proteins were associated with AD status or t-tau concentration at a significance below of 0.05 in each plot.

Coefficients of variation (CVs) were calculated in RStudio for all inter-plate and intra-plate combinations of three samples, the median thereof was reported as overall coefficient of variation. Combinations with only one observation in three samples of a given protein were excluded. The protein CVs of the main study were calculated likewise within cohorts individually. The median CVs were calculated within the three cohorts, and the median thereof reported as final CV.

### Machine learning for participant classification

All data processing was done in Python (3.7.3). Protein intensity data were Z-scored within cohorts, saved in Excel, and imported via the pandas package (0.25.3). Except for the XGBoost classifier, missing intensities were replaced with 0. Machine learning classifiers were employed using the scikit-learn package (0.21.3) and the XGBoost package package (0.90) (Fabian et al, 2011). Results were plotted via matplotlib (3.1.2). Visualization of the decision tree was performed with the dtreeviz package (https://github.com/parrt/dtreeviz).

In order to estimate features important for AD prediction, we employed a decision tree (Freund & Schapire, 1997). The minimum depth of the tree was increased until a training accuracy of 1.0 was achieved. At a tree depth of 2, using the protein intensities of MAPT, PKM (protein group P14618-2), and MIF, the training accuracy had reached 0.86, highlighting the importance of these proteins for the classifier. For a tree depth of six, intensities of a total of 14 proteins were used by the algorithm.

For estimating how well our tree-based approach would generalize to new data, we tested several ensemble methods (AdaBoost, Bagging, ExtraTrees, GradientBoosting, RandomForests, XGBoost). The subset of 14 protein intensities selected by the decision tree above were randomly shuffled and split using a k-Folds cross-validator ($k = 6$). Each model was used with its default parameters. XGBoost had the best performance and was selected for further analysis. To determine the optimum set of features, we added proteins to the model iteratively according to their feature importance within the tree (Fig 5A) and compared the AUC as a measure of model performance. To control for overfitting, we employed early stopping with 10 rounds and logloss as evaluation metric for best generalizability. No further tuning of hyperparameters was performed at this stage.

To assess the sensitivity and specificity of the final method, we combined each train and test set of the cross-validation and calculated the confusion matrix. Here, a test accuracy of 0.85 was achieved (training accuracy 0.94; sensitivity 82%, specificity 87% on our AD data).

## Data availability

The datasets produced in this study are available in the following databases:

- Proteomic datasets: PRIDE archive PXD016278 (Perez-Riverol et al, 2019; https://www.ebi.ac.uk/pride/archive/)

**Expanded View** for this article is available online.

## Acknowledgements

We thank all members of the Proteomics and Signal Transduction Group at the Max Planck Institute of Biochemistry and the Clinical Proteomics Group at the NNF Center for Protein Research for help and discussions and in particular Igor Paron, Christian Deiml, and Alexander Strasser for technical assistance. We further thank Martin Steger for his contribution in establishing data-independent mass spectrometry for CSF and Sebastian Virreira Winter, Özge Karayel, Niels H. Skotte, Felix Meissner, and Daniel Hornburg for discussions and supplying samples. The work carried out in this project was partially supported by the Max Planck Society for the Advancement of Science, the European Union's Horizon 2020 research and innovation program with the Microb-Predict project (no. 825694), by grants from the Novo Nordisk Foundation (NNF15CC0001; NNF15OC0016692), the DFG project "Chemical proteomics inside us" (no. 412136960), the European Research Council Synergy Grant under FP7 GA number ERC-2012-SyG_318987-Toxic Protein Aggregation in Neurodegeneration (ToPAG), and funding from the Harvard T.H. Chan School of Public Health Dean's Challenge Program sponsored by the McLennan Family Fund.

## Author contributions

JMB, PEG, and JBM designed the experiments, performed, analyzed, and interpreted the MS-based proteomic data. JMB optimized the MS data processing, performed bioinformatics analysis, and generated text and figures for the manuscript, and PEG contributed to these aspects. MTS performed the machine learning analysis. MK, FL, PK, DB, CGS, EII, OP, ND, MS, MKJ, and HZ designed and established the study cohorts and contributed to writing the paper. MM supervised and guided the project, designed the experiments and interpreted MS-based proteomics data and wrote the manuscript.

## Conflict of interest

The authors declare that they have no conflict of interest.

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
