## [Review Process File · Molecular Systems Biology]

Proteome Profiling in Cerebrospinal Fluid Reveals Novel Biomarkers of Alzheimer's Disease

Matthias Mann, Jakob Bader, Philipp Geyer, Johannes Müller, Manja Koch, Maximilian Strauss, Frank Leypoldt, Peter Koertvelyessy, Daniel Bittner, Carola Schipke, Enise Incesoy, Oliver Peters, Nikolaus Deigendesch, Mikael Simons, Majken Jensen, and Henrik Zetterberg

DOI: [10.15252/msb.20199356](https://doi.org/10.15252/msb.20199356)

Corresponding author: Matthias Mann (mmann@biochem.mpg.de)

Review Timeline:

Submission Date:	14th Nov 19
Editorial Decision:	17th Dec 19
Revision Received:	31st Mar 20
Editorial Decision:	23rd Apr 20
Revision Received:	29th Apr 20
Accepted:	30th Apr 20

Editor: Maria Polychronidou

Transaction Report:

17th Dec 2019

Manuscript Number: MSB-19-9356, Proteome Profiling in Cerebrospinal Fluid Reveals Novel Biomarkers of Alzheimer's Disease

Thank you again for submitting your work to Molecular Systems Biology. We have now heard back from the three referees who agreed to evaluate your study. As you will see below, the reviewers acknowledge that the study seems interesting in the context of biomarker discovery for AD. They raise however a series of concerns, which we would ask you to address in a major revision.

Without repeating all the points listed below, the most fundamental issue raised refers to the differences between the cohorts, which makes the comparison between them somewhat complicated. The differences need to be taken into account more thoroughly during the analyses. Both reviewers #1 and #2 point out that the re-classification of the Berlin cohort using the proteomic results seems problematic and it would be better to re-classify the cohort beforehand. The concern regarding the differences between cohorts is echoed by reviewer #3, who, besides this comment, unfortunately does not make particularly constructive suggestions on how to improve the study. Reviewers #1 and #2 also mention that better statistical support needs to be provided for the presented biomarkers.

All other issues raised by the reviewers would need to be convincingly addressed. Please let me know in case you would like to discuss in further detail any of the issues raised and how you plan to address them.

On a more editorial level, we would ask you to address the following issues:

- Please provide a .doc file for the main text.
- We have replaced Supplementary Information by the Expanded View (EV format). In this case, all additional figures can be included in a PDF called Appendix. Appendix figures should be labeled and called out as: "Appendix Figure S1, Appendix Figure S2..." etc. Each legend should be below the corresponding Figure/Table in the Appendix. Please include a Table of Contents in the beginning of the Appendix. For detailed instructions regarding expanded view please refer to our Author Guidelines: .
- Tables EV1-EV5 need to be provided as Datasets EV1-EV5. Please provide them as individual .xls files, each one containing a short description of the dataset in a separate tab.
- The citation to the preprint by Higginbotham et al, needs to be removed from the abstract. It is of course fine to include the citation in the main text.
- All Materials and Methods need to be described in the main text. We would encourage you to use

'Structured Methods', our new Materials and Methods format. According to this format, the Material and Methods section should include a Reagents and Tools Table (listing key reagents, experimental models, software and relevant equipment and including their sources and relevant identifiers) followed by a Methods and Protocols section in which we encourage the authors to describe their methods using a step-by-step protocol format with bullet points, to facilitate the adoption of the methodologies across labs. More information on how to adhere to this format as well as downloadable templates (.doc or .xls) for the Reagents and Tools Table can be found in our author guidelines: . An example of a Method paper with Structured Methods can be found here: .

- Please include a Data availability section describing how the data (and code) generated in this study have been made available. This section needs to be formatted according to the example below:

The datasets and computer code produced in this study are available in the following databases:

- Chip-Seq data: Gene Expression Omnibus GSE46748

(<https://www.ncbi.nlm.nih.gov/geo/query/acc.cgi?acc=GSE46748>)

- [data type]: [full name of the resource] [accession number/identifier] ([doi or URL or identifiers.org/DATABASE:ACCESSION])

- Please provide a "standfirst text" summarizing the study in one or two sentences (approximately 250 characters), three to four "bullet points" highlighting the main findings and a "synopsis image" (550px width and max 400px height, jpeg format) to highlight the paper on our homepage.

- When you resubmit your manuscript, please download our CHECKLIST

(<http://bit.ly/EMBOPressAuthorChecklist>) and include the completed form in your submission.

Please note that the Author Checklist will be published alongside the paper as part of the transparent process

(<https://www.embopress.org/page/journal/17444292/authorguide#transparentprocess>).

If you feel you can satisfactorily deal with these points and those listed by the referees, you may wish to submit a revised version of your manuscript. Please attach a covering letter giving details of the way in which you have handled each of the points raised by the referees. A revised manuscript will be once again subject to review and you probably understand that we can give you no guarantee at this stage that the eventual outcome will be favorable.

If you do choose to resubmit, please click on the link below to submit the revision online *within 90 days*.

Link Not Available

IMPORTANT: When you send your revision, we will require the following items:

1. the manuscript text in LaTeX, RTF or MS Word format
2. a letter with a detailed description of the changes made in response to the referees. Please specify clearly the exact places in the text (pages and paragraphs) where each change has been made in response to each specific comment given
3. three to four 'bullet points' highlighting the main findings of your study
4. a short 'blurb' text summarizing in two sentences the study (max. 250 characters)
5. a 'thumbnail image' (width=211 x height=157 pixels, Illustrator, PowerPoint, OmniGraffle or jpeg format), which can be used as 'visual title' for the synopsis section of your paper.
6. Please include an author contributions statement after the Acknowledgements section (see <https://www.embopress.org/page/journal/17444292/authorguide>)
7. Please complete the CHECKLIST available at (<http://bit.ly/EMBOPressAuthorChecklist>). Please note that the Author Checklist will be published alongside the paper as part of the transparent process (<https://www.embopress.org/page/journal/17444292/authorguide#transparentprocess>).
8. Please note that corresponding authors are required to supply an ORCID ID for their name upon submission of a revised manuscript (EMBO Press signed a joint statement to encourage ORCID adoption). (<https://www.embopress.org/page/journal/17444292/authorguide#editorialprocess>)

Currently, our records indicate that the ORCID for your account is 0000-0003-1292-4799.

Link Not Available

The system will prompt you to fill in your funding and payment information. This will allow Wiley to send you a quote for the article processing charge (APC) in case of acceptance. This quote takes into account any reduction or fee waivers that you may be eligible for. Authors do not need to pay any fees before their manuscript is accepted and transferred to the publisher.

*** PLEASE NOTE *** As part of the EMBO Press transparent editorial process initiative (see our Editorial at <http://dx.doi.org/10.1038/msb.2010.72>), Molecular Systems Biology publishes online a Review Process File with each accepted manuscripts. This file will be published in conjunction with your paper and will include the anonymous referee reports, your point-by-point response and all pertinent correspondence relating to the manuscript. If you do NOT want this File to be published, please inform the editorial office at msb@embo.org within 14 days upon receipt of the present letter.

Reviewer #1:

The manuscript by Bader et al. describes a proteomic profiling by advanced mass spectrometry of small amounts of CSF from 3 different study cohorts to find biomarkers for Alzheimer's disease. Each study obtained CSF from approximately 30 AD and 30 control subjects for a total of over 200

subjects. The mass spectrometric analysis was well done, with deep proteomic coverage of over 1000 proteins and low CV (usually <20%), and the strategy of using similar sample sizes for biomarker candidate discovery and verification is interesting and promising. Known indicators of neurodegeneration including tau, SOD1 and PARK7 were leading biomarker candidates found in addition to others to construct a core panel of 31 potential biomarkers that includes several proteins suggestive of leakage from damaged brains. However, the 3 separate studies that are compared have several differences in the way AD subject populations were selected, so comparison of the results between studies is problematic and complicated- the main problem being the unreliability of the assignment of subjects to AD and controls, culminating in the need to reclassify the Berlin group based on proteomics data. Overall, the analytical aspects of the strategy for biomarker discovery and validation are promising, but its application to AD is difficult to assess because of the classification/diagnosis difficulties currently inherent in the AD field.

Specific comments:

1. One problem with the study is that the 3 separate cohorts were chosen based on different criteria. For the Sweden study, subjects with cognitive impairment at a memory clinic with total tau >400 ng/L, phospho-tau >60 ng/L (no phosphosites specified), and Abeta < 550 ng/L were placed into the AD group, while "none of the control samples fulfilled these criteria". Does this mean they fulfilled none of the criteria or did not fulfill all of the criteria (i.e. if 2/3 are fulfilled are they still eligible for the control group?). For the Magdeburg cohort AD subjects were classified based on clinical evaluation of cognitive function (CERAD neuropsychological test battery) and MRI, in addition to one of the following: either p-tau181 > 80 ng/L AND t-tau > 450 ng/L, OR Abeta1-42 < 485 ng/L AND Abeta1-42/Abeta1-40 ratio < 0.6. Controls had no evidence of neurodegeneration and had normal cell counts, protein amounts, and lactate concentration in CSF. Kiel control subjects presented with acute headache but no history of dementia, systemic or CNS inflammation, or blood-brain barrier dysfunction. These controls were significantly younger than any of the AD populations (median age 32 versus 70 for other populations), which is a serious problem for any AD/non-AD comparison. The Berlin AD CSF came from AD patients diagnosed on the basis of cognitive and functional tests including American Psychiatric Association clinical guidelines, DSM-5 and MMSE-scoring, and cMRI, and had Abeta < 600 ng/L or Abeta1-42/Abeta1-40 ratio < 0.6 in addition to t-tau > 350 ng/L. For some reason, perhaps related to these differences, the classification of the Berlin cohort did not agree with the other 2 cohorts according to the proteomics results.

2. Results page 4: were these biochemical measurements for tau and Abeta performed in different labs at the points of diagnosis, or on CSF samples by the authors? I believe the former, and given that slightly different methods were used for the measurements, comparison between cohorts is problematic.

3. Page 6, proteomic differences: The fact that approximately 500 proteins out of an average of 1230 identified proteins change "significantly" is surprising to me. How is it useful or informative when so many proteins change? Perhaps it might have been more interesting to use multiple hypothesis-corrected p-values to narrow down this list and increase statistical significance. Also, in Figure 2A it seems that many more proteins are more abundant in the AD Swedish cohort than less abundant- how is this possible if total protein amounts were normalized in the AD and control samples? Also, since tau levels were a criterion for assignment of patients to the AD groups, why should it be surprising or noteworthy that tau was a highly changing protein between AD and controls?

4. page 7, second paragraph: Berlin AD patients were required to have >350 ng/L t-tau, but the other 2 cohorts required >400 ng/L t-tau (Sweden) or > 450 ng/L t-tau (Magdeburg). Could this explain some of the differences in number of differentially expressed proteins? This assumes that t-tau levels correlate with other protein changes- not unlikely given that the protein ratio cutoff is 1.3.

5. Bottom of page 7: It is not at all remarkable that 191 proteins changed in both Magdeburg and Sweden cohorts. If out of 1200 proteins 500 are changed in each, 191 seems to be a virtually random overlap between the 2 groups of 500. It would be much more interesting to see if the percent overlap is much better than approximately 40% (191/approximately 500) if the statistical requirements for change were more stringent than uncorrected p-value of < 0.05 Also, was the direction of change required to be consistent to qualify as overlapping? It should be.

6. page 8, first paragraph: It is worrying that of the 37 proteins differing "significantly" by AD status in all 3 cohorts, only 28 were in the same direction. To me this indicates a problem with the overall dataset. Narrowing the panel to tau-associated proteins now biases the proteomic assay to samples that have increased tau, which was a prerequisite for inclusion in the AD groups. The proteomic assay would then simply reinforce the ELISA tau measurements made at the point of diagnosis, rather than indicate AD per se.

7. Page 8, 3rd paragraph, last sentence: is this about one patient or more? "The second-most AD-like participants... suggesting that they had an AD-like phenotype". Please clarify or correct.

The reclassification of the Berlin group based on proteomic signatures implies that the original diagnoses of AD vs control were faulty. By definition the new Berlin groups would behave better in the proteomic assays as shown in Fig 3. But what about the clinical status of the rest of the Berlin cohort in addition to the top 2 most AD-like subjects? Did these statuses agree with the proteomics predictions? Also, would the original classification of the Berlin cohort look more like the reclassified cohort if more than 350 ng/L t-tau have been required?

8. Page 11, proteins differing by AD status correlate with CSF t-tau abundance: Again, this should be expected since the AD cohorts were largely assigned based on t-tau measurements by ELISA.

9. page 13, bottom: Actually, of 108 proteins, 42 significant proteins in agreement with the Higginbotham study (even fewer with directional agreement, which are the only proteins that should count) does not seem like good agreement- this suggests that the majority of biomarker candidates are not confirmed. Therefore the sentence about high correlation at the top of page 14 should be toned down a bit. Would percent agreement between the 2 studies increase if inclusion criteria (lower p-value, or use of q-value or corrected p-value) were more stringent?

Minor point

10. page 13, 2nd paragraph line 3: should be "differed significantly.."

Reviewer #2:

In this study, Bader et al. performed unbiased proteomic analysis of a total 208 AD and non-AD CSF samples derived from three separate cohorts. These cohorts were notably stratified into non-AD and AD groups in varied ways. While the Sweden cohort exclusively used biochemical CSF levels of

Abeta and tau to diagnostically classify their cases, the Magdeburg / Kiel and Berlin cohorts used a combination of clinical presentation and biochemical data. Also notable, the Sweden and Magdeburg / Kiel biochemical measures (t-tau, p-tau181, Abeta1-42) differed from those of the Berlin group (t-tau, Abeta1-40, Abeta1-42).

Using a streamlined automated approach for sample digestion, the 208 samples were analyzed via DIA mass spectrometry, resulting in the quantification of an average 1200 proteins per sample. High inter-assay reproducibility was achieved for over 1000 of the proteins (CV < 20%). The authors then used a clustering analysis to examine functional relationships between the quantified proteins, identifying a relatively large synapse-associated MAPT-containing protein group. Differential expression analysis of quantified proteins revealed robust levels of differential protein expression ($p < 0.05$) between the AD and non-AD groups of the Sweden and Magdeburg / Kiel cohorts (~400 to 500 proteins). Most of these altered proteins were increased in AD CSF and enriched with neuron-associated gene ontologies. However, the Berlin group featured a markedly lower number of differentially expressed proteins between its AD and non-AD subjects (~100 proteins). The authors linked this result to the poor biochemical (i.e. AD biomarker) separation the Berlin cohort demonstrated in its clinical amyloid and tau assays. MAPT and YWHAG were two of the most significantly increased proteins in AD among all three cohorts.

The authors found consistent protein changes across the Sweden and Magdeburg / Kiel cohorts (~200 proteins). However, when looking for consistent protein changes across all three cohorts, only 37 proteins were identified. The authors then narrowed these 37 proteins to an 11-protein panel by including only those found in the tau-containing cluster of their initial global correlation map. These 11 proteins were then used to re-classify their Berlin cohort into control-like, intermediate, and AD-like clusters. The authors then looked for consistent protein changes across the three cohorts using this re-classified Berlin cohort, excluding the intermediate group. This produced a panel of 130 proteins, which they narrowed to 31 proteins based on consistent changes within an independent CSF analysis performed by Higginbotham et al. Machine learning was then used to further narrow these 31 proteins to 13 CSF protein signatures of AD.

Overall, this study was interesting and highlighted advancements in high-throughput proteomic methods for CSF analysis with regards to proteome depth and reproducibility. The authors should also be commended for comparing their AD biomarker findings with recently deposited pre-prints as this enhances the rigor and reproducibility of their results. However, there are several major limitations with interpretations of the results mainly involving the rubric used for biomarker prioritization that need to be addressed, prior to publication.

Major Concerns

- First, the authors' rubric for marker prioritization in the Berlin cohort is disturbingly circular. As presented, the Berlin cohort was re-classified based on the discovery proteomic results, essentially forcing it into proteomic consistency with the other two cohorts. Then, consistency across these three cohorts was used as a criterion to prioritize biomarkers following this re-classification. The authors justify this circular rubric using scant anecdotal evidence and suggests that several individuals in this Berlin cohort were in the presymptomatic stages of AD and simply misclassified at the outset. However, should not this presumption be further substantiated? There are established research guidelines for the classification of presymptomatic AD based on cognitive and core biomarker status. Would it not make more sense-and generate a much less convoluted rubric-to re-stratify the Berlin individuals based on these guidelines? For instance, Tau/Abeta ratios are generally accepted as the most accurate way of biochemically defining cognitive normal individuals

with presymptomatic disease. Yet, such ratios were not mentioned in the results of the manuscript. The criteria used to determine cognitive status was also not clearly defined in the results. It is possible stricter MMSE cut-offs could have better stratified this Berlin group as well, prior to proteomic analysis? In summary, the authors should eliminate the circular proteomic re-classification of the Berlin group and explore defining them based on more rigorous clinical and core biomarker research criteria used in the field or exclude the cohort entirely as the re-classification confounds the AD biomarker findings.

- A second concern generated by the proteomic re-classification of the Berlin group is the newly created presence of an "intermediate" sub-group. This group included a third of the total Berlin cohort, including 15 individuals originally classified as AD. This intermediate group was promptly identified and eliminated from analysis in the second half of the manuscript. Yet, there was no discussion as to what these "intermediate" individuals signify. Do the authors consider all these cases presymptomatic AD? Or are they more consistent with MCI or mild AD? Could a portion of these intermediate individuals represent a degree of biological heterogeneity among true AD cases? This is simply another unclear / unsatisfying aspect of the circular re-classification of the Berlin group.

- Third, perhaps due to the issues outlined above, the overall scheme of the paper was confusing and difficult to follow. When the authors mentioned a new protein panel of interest, I often had to revisit previous paragraphs to determine the panel's derivation. Could the process of marker prioritization perhaps be more streamlined? A global depiction of the algorithm would be helpful.

Minor Concerns

- Pg 4. The following sentence needs to be reworded, as it is awkward and the numbers do not compute: "Accordingly, we collected and analyzed three separate study populations each comprised around 30 AD subjects and 30 non-AD controls, totaling 208 individuals."

- Pg 4. The precise clinical and biochemical cut-offs to classify non-AD and AD individuals in the three cohorts should be explicitly stated.

- Pg. 4. The non-AD sub-group of the Magdeburg/Keil cohort was substantially younger than the AD group. What effect does this have on the proteomic results? Were age and sex regressed as co-variables in the bioinformatic workflow? The authors should discuss and justify.

- Pg 5. The authors state "CSF has much less protein than plasma". This statement should be followed by a citation. Also, how much less protein? Has this been quantified? Was more volume of CSF loaded on the instruments to get equivalent signal?

Pg. 5. The authors highlighted the second-largest cluster of their original unbiased global correlation map and used it later in the study to prioritize markers of interest. They in part justify this stating that due to its functional annotations (neuronal, axonal, and synaptic), this cluster reflects "the brain contribution to the CSF proteome". However, this statement is somewhat presumptuous. Is this cluster more enriched than the others with proteins detectable in the brain? Do the other clusters not contain any proteins detectable in brain? The authors also imply that the other clusters represent some sort of contamination or that their presence is related to "quality issues". However, where is the justification of this? The pre-print paper they reference (Higginbotham et al.) demonstrated consistent disease-related CSF changes in plasma / vascular-derived proteins. How is this to be explained if such clusters are only related to technical issues?

- Pg 6. The authors state that "...we and others had altogether failed to detect tau or at significantly changed levels in proteomic studies". This is not true, as the pre-print paper the authors referenced (Higginbotham et al.) detected changes in tau in CSF1 as well as the published findings from the Pandey group (PMID: 30578620). Several PRM (not dissimilar to DIA-MS) studies have also detected and quantified Tau (J. Proteome Res. 2016, 15, 2, 667-676).
- Pg 12. The authors state that "glycolytic proteins may originate from astrocytes...". The authors should cite the pre-print they cite in their abstract (Johnson et al.,) as this paper links the metabolic/glycolytic biomarkers in CSF to differences in these same markers in AD brain.
- Pg. 13. The authors should revisit and download the protein fold change information for CSF1 in BioRxiv pre-print by Higginbotham et al. My review of this data shows an increase their number of validated proteins increases from 31 to 38. For instance, the fold changes for CHI3L1, SOD2, and STMN1 are all increased in AD, whereas SERPINF2, KLKB1, FETUB, and APOA2 are all decreased, further validating the authors' results.
- P. 5. "1484 proteins much higher than reported in published literature." This is not correct. There are a number of studies with >2000 proteins identified in CSF by MS (see Pandey publication and the Higginbotham preprint above). Maybe the authors are referring to the number of proteins contributing to protein quantified in CSF without albumin depletion by DIA-MS or the number quantified at a given missingness threshold? It's not clear from this sentence.
- The clustering section is highly confusing. If clustering is performed using the 37 proteins that are significantly different between AD and controls among all three cohorts, wouldn't one expect that AD cases in each cohort would cluster together? And why did only 27 of the 37 have higher abundance across the 3 cohorts after this analysis? Also, they could have used these 27 or 37 proteins to re-classify the Berlin cohort. Why go down to the 11 in the tau cluster? Justification is not provided.
- The authors shouldn't speculate that the abundance calculations in brain equate to protein levels in CSF and reflect "leakage" or tissue damage. That may be the case for some proteins, but not all. There is likely exosome release and other mechanisms underlying the levels in brain and CSF.
- Fig. 4D, legend states "inversely correlating" with tau. Looks like regular correlation.
- P. 13, referring to glycolytic signature, the authors should cite Johnson et al. pre-print and other previous studies in the discussion when referring to glycolytic proteins elevated in CSF.
- Expanded View Fig 2 C-E legend labels are incorrect
- Supplemental Tables are not well formatted and difficult read.

Reviewer #3:

This work is extremely important work and we need new biomarkers and hypotheses on why we get AD. However, this is half finished. The findings needs to be validated by another technique, the days have gone where you run a whole of samples and come to a conclusion based on that because omics often kicks up red herrings. These red herrings can send researchers down a blind

ally. Also the reason why there is centre to centre differences haven't been addressed properly.

Point-by-point answers of reviewer's comments

We thank the Reviewers for the in-depth and insightful comments on our manuscript “**Proteome Profiling in Cerebrospinal Fluid Reveals Novel Biomarkers of Alzheimer's Disease**”

In summary, the Reviewers appreciate that our study breaks new ground in several respects such as an advanced cerebrospinal fluid (CSF) proteomics workflow and an innovative study design approach combining multiple medium-sized cohorts. They appreciate that the proteomics workflow is characterized by high proteomic depth and reproducibility while still being amenable to high throughput. Furthermore, Alzheimer's disease (AD) is recognized as an extremely important field of application in search of better biomarkers. Our approach clearly identified known markers of AD providing positive controls and we present a core panel of more than 20 CSF proteins linked by strong evidence to AD. Reviewers 1 and 2 had a common concern about the re-classification of the Berlin cohort, which we have addressed as summarized just below. Reviewer 3 finds this to be an ‘extremely important work’, and their concerns about ‘centre to centre differences’, should also be addressed with unified criteria described below.

Consistent up-front classification based on clinical laboratory CSF AD markers:

Reviewers 1 and 2 raised the following two main concerns: i) they found the proteomics-based reclassification of the Berlin cohort circular and ii) they questioned the differences across cohorts with regards to the AD classification criteria. They point out that AD patients were defined differently in the three cohorts and that the control populations varied as well. However, as Reviewer 1 also remarked, these difficulties are currently inherent to the AD field. As there is no universally accepted standard for the diagnosis of AD in clinical practice, multi-centric cohorts will differ in selection criteria.

Given the difficulties in obtaining AD CSF populations and especially controls, we are quite satisfied with the size and quality of the three cohorts. However, we do agree with concerns of the reviewers and have fundamentally changed our analyses as a result. We now apply a uniform classification up front to the three cohorts based on ELISA measurements for the same clinical AD biomarkers in CSF (t-tau, A β ₁₋₄₂, A β ₁₋₄₀) and eliminate the proteomics-based reclassification of the Berlin cohort.

We had initially reclassified the Berlin cohort based on eleven proteins and subsequently compared the resulting AD-like and non-AD-like clusters regarding their entire quantified proteomes (~1200 proteins). To Reviewer 2 this appeared to be circular with regards to the proteomics-based stratification. Following Reviewer 2's suggestion, we now exclude the proteomics-based reclassification and classify all cohorts (including the Berlin cohort) based on well-established AD CSF biomarkers: Patients were classified as AD if the t-tau concentration

was above 400 ng/L and at least one of the A β criteria fulfilled, including A β_{1-42} < 550 ng/L or A β_{1-42} /A β_{1-40} ratio < 0.065. This stringent and uniform classification, stipulating both a t-tau and an A β criterion, results in an AD group with as few false positives as possible. We also compared this stringent classification to tau/A β ratios as suggested by Reviewer 2, which demonstrates that the former is more stringent than the latter and results in similar but slightly higher cross-cohort consistency regarding the proteomics results.

With the consistent classification scheme based on clinical AD CSF values, several samples in the Magdeburg and Berlin cohorts were excluded due to missing clinical AD CSF biomarker values, reducing the overall number of samples from 208 to 197. The new AD/non-AD classification is very consistent with the previous cohort-specific classifications for the Sweden and Magdeburg cohorts (Review Fig R1). In the Berlin cohort, the classifications are also largely consistent but 15% of the previous non-AD controls are now classified as AD, leading to much better agreement between this and the other two cohorts.

		previous cohort-specific classification					
		Sweden		Magdeburg		Berlin	
		AD	non-AD	AD	non-AD	AD	non-AD
new uniform classification	AD	29	0	25	1	24	9
	non-AD	1	30	0	12	0	50
	excluded	0	0	3	1	6	1

Review Figure R1: Previous vs. new AD/non-AD classification. Numbers indicate the number of participants with a given status (AD/non-AD) in each cohort. Green indicates agreement between the previous and current classification, whereas grey indicates differences.

The standardized classification suggested by the reviewers indeed improved consistency and comparability across all cohorts and increased the observed effect sizes, compared to the classification obtained from the clinical centers. Specifically, the unified biochemical criteria for AD increased the AD-related CSF proteome differences in the Berlin cohort to a level more similar to the other two cohorts. Compared to the previous cohort-specific AD classification, the number of significant proteins in the Berlin cohort increased from 108 to 168 (+55%) and from 4 to 22 (+450%) at p-value and q-value cutoffs of 5%, respectively. Likewise the three-cohort agreement – the number of significant ($p < 5\%$) proteins with consistent AD-regulation directionality – increased from 28 of 37 (76%) to 40 of 43 (93%), respectively.

Moreover, quantitative agreement of AD/non-AD fold changes for significant proteins between the Berlin cohort and the two other cohorts increased drastically from Pearson's coefficients of 0.39 to 0.74 (vs Sweden) and 0.54 to 0.80 (vs Magdeburg/Kiel). Furthermore, the increased agreement between cohorts of this study also translates to a 100% directionality agreement of significant proteins with the CSF1 dataset of the Higginbotham et al study. We conclude that the revised classification, is consistent, largely preserves the previous one, with some changes for the Berlin cohort, and substantially increases qualitative and quantitative agreement across cohorts. It is therefore the basis for the entire revised manuscript.

Reviewer 1

The manuscript by Bader et al. describes a proteomic profiling by advanced mass spectrometry of small amounts of CSF from 3 different study cohorts to find biomarkers for Alzheimer's disease. Each study obtained CSF from approximately 30 AD and 30 control subjects for a total of over 200 subjects. The mass spectrometric analysis was well done, with deep proteomic coverage of over 1000 proteins and low CV (usually <20%), and the strategy of using similar sample sizes for biomarker candidate discovery and verification is interesting and promising. Known indicators of neurodegeneration including tau, SOD1 and PARK7 were leading biomarker candidates found in addition to others to construct a core panel of 31 potential biomarkers that includes several proteins suggestive of leakage from damaged brains. However, the 3 separate studies that are compared have several differences in the way AD subject populations were selected, so comparison of the results between studies is problematic and complicated- the main problem being the unreliability of the assignment of subjects to AD and controls, culminating in the need to reclassify the Berlin group based on proteomics data. Overall, the analytical aspects of the strategy for biomarker discovery and validation are promising, but its application to AD is difficult to assess because of the classification/diagnosis difficulties currently inherent in the AD field.

We thank the reviewer for the overall positive and detailed evaluation of our work, especially with regards to the overall study design.

Major points

1. One problem with the study is that the 3 separate cohorts were chosen based on different criteria. For the Sweden study, subjects with cognitive impairment at a memory clinic with total tau >400 ng/L, phospho-tau >60 ng/L (no phosphosites specified), and Abeta < 550 ng/L were placed into the AD group, while "none of the control samples fulfilled these criteria". Does this mean they fulfilled none of the criteria or did not fulfill all of the criteria (i.e. if 2/3 are fulfilled are they still eligible for the control group?). For the Magdeburg cohort AD subjects were classified based on clinical evaluation of cognitive function (CERAD neuropsychological test battery) and MRI, in addition to one of the following: either p-tau181 > 80 ng/L AND t-tau > 450 ng/L, OR Abeta1-42 < 485 ng/L AND Abeta1-42/Abeta1-40 ratio < 0.6.

Controls had no evidence of neurodegeneration and had normal cell counts, protein amounts, and lactate concentration in CSF. Kiel control subjects presented with acute headache but no history

of dementia, systemic or CNS inflammation, or blood-brain barrier dysfunction. These controls were significantly younger than any of the AD populations (median age 32 versus 70 for other populations), which is a serious problem for any AD/non-AD comparison. The Berlin AD CSF came from AD patients diagnosed on the basis of cognitive and functional tests including American Psychiatric Association clinical guidelines, DSM-5 and MMSE-scoring, and cMRI, and had Abeta < 600 ng/L or Abeta1-42/Abeta1-40 ratio < 0.6 in addition to t-tau > 350 ng/L. For some reason, perhaps related to these differences, the classification of the Berlin cohort did not agree with the other 2 cohorts according to the proteomics results.

We agree with the reviewer that the classifications provided by the different clinical centers are not easily harmonized post-hoc, as we had tried initially. Following this and Reviewer 2's advice, we now follow a uniform AD classification as outlined in the summary above and described in detail in the Materials and Methods section. As noted above, this reclassification was beneficial in every way, including much better agreement of the Berlin cohort with the others as outlined in the summary above. However, there is still a larger fraction of samples in the grey zone between AD and non-AD than the other cohorts as exemplified by the limited separation regarding clinical AD CSF biomarkers (Fig EV1G-K). As a result, our proteomic results reflect less separation and smaller effect sizes between AD and non-AD samples in the Berlin cohort. Nevertheless, despite the still reduced effect size, qualitative and quantitative cross-cohort agreement of proteins that are significant in all three cohorts is excellent (93% overlap) with the new uniform clinical AD CSF biomarker-based classification.

Again, we agree with the concern regarding the Kiel controls. The Magdeburg/Kiel cohort comprises two non-AD control entities, namely the Magdeburg and Kiel controls and we acknowledge that the Kiel control group (16 samples and 8% of total samples) is not ideal because of the indications that the Reviewer lists. To investigate if these differences have any material influence, we compared outcomes of the analysis with each control group separately. Overall, our revised analysis led to a panel of 40 proteins that are statistically significant and have consistent directionality (new Fig. 3F). AD/non-AD fold changes of these 40 proteins comparing Magdeburg AD to either single non-AD control group have a 100% consistency in directionality and are quantitatively highly correlated (Pearson's correlation coefficient $r = 0.81$, Fig EV3B). Combined with the fact that the 40 proteins are consistent in all our cohorts, we conclude that the Kiel control samples do not skew the results. Furthermore, to specifically test whether the age difference of the Kiel controls – and the sex difference pointed out by Reviewer 2 – had any influence on our conclusions, we performed a regression analysis, with these two covariates. This confirmed that that AD status is still a significant ($p < 0.05$) predictor for the CSF abundance of these 40 proteins (Fig EV3C).

The precise biochemical thresholds and the integration of these multiple cutoffs for classification is stated in the revised Materials and Methods section as follows:

“Patients were classified as AD if the t-tau concentration was above 400 ng/L and the $A\beta_{1-42}$ concentration below 550 ng/L or the $A\beta_{1-42}/A\beta_{1-40}$ ratio was below 0.065. The t-tau criterion and at least one of the two $A\beta$ criteria had to be met for a patient to be classified as having AD and patients were classified as not having AD otherwise.”

2. Results page 4: were these biochemical measurements for tau and Abeta performed in different labs at the points of diagnosis, or on CSF samples by the authors? I believe the former, and given that slightly different methods were used for the measurements, comparison between cohorts is problematic.

The Reviewer is correct: the biochemical measurements for tau and A β were performed at the sites of CSF collection. We have now made this explicit in the Materials and Methods section. However, these assays were performed in clinical laboratories as routine analysis underlying stringent reproducibility standards with the aim of absolute quantification for clinical diagnosis. Moreover, the same kits sourced from the company Fujirebio were used for both tau and A β quantitation at all clinical sites with the only exception that in Berlin a kit from Meso Scale Diagnostics was used for A β . Such clinical tests for absolute quantification are usually considered reliable regardless of the site where they are performed.

3. Page 6, proteomic differences: The fact that approximately 500 proteins out of an average of 1230 identified proteins change "significantly" is surprising to me. How is it useful or informative when so many proteins change? Perhaps it might have been more interesting to use multiple hypothesis-corrected p-values to narrow down this list and increase statistical significance.

In clinical settings, a small set of clinical AD CSF biomarkers is used for assigning disease status; generally only t-tau, p-tau₁₈₁, A β ₁₋₄₂, and A β ₁₋₄₀. It is in light of this low number that we find the large-scale alterations 'surprising'. However, given the atrophy that is apparent in AD upon autopsy, it is perhaps expected that the concentration of many proteins is changing in AD CSF. We point this out in the revised manuscript:

“The extensive brain atrophy apparent upon autopsy and the widespread brain proteome alterations harmonize well with the observed substantial alterations in the CSF proteome in AD and other neurodegenerative diseases (Higginbotham *et al*, 2019; Hosp *et al*, 2017)”.

We agree with the Reviewer that a set of 500 proteins (from the Swedish cohort) is impractical for biomarker purposes and have used several ways to narrow it down. After requiring consistency in all cohorts and multiple-hypothesis testing, as suggested by the reviewer, we end up with 12 proteins, which are biologically plausible. This is indeed a much more manageable set for future biomarker applications. In the revised manuscript, we discuss both the set of 40 proteins that is significant and common across the cohorts and this set of 12 proteins.

Also, in Figure 2A it seems that many more proteins are more abundant in the AD Swedish cohort than less abundant- how is this possible if total protein amounts were normalized in the AD and control samples?

We thank the reviewer for highlighting this point. Regarding the total protein amount, it is correct that this is normalized. Firstly, for each sample 500 ng of peptide – as judged by Nanodrop concentration measurement (absorbance at 280 nm see Material and Methods) - is loaded onto the chromatography column for LC-MS/MS analysis. Secondly, the processing software normalizes the proteomes of each sample *in silico*, for this we used the ‘Local

Normalization algorithm' in the Spectronaut software which is based on a local regression normalization model (Callister *et al*, 2006). However, normalization of each sample does not entail that equal numbers of proteins change in either direction of a biological comparison (AD vs non-AD). Indeed, prompted by the reviewer, we analyzed the overall fold-change distributions and found that they were bell-shaped but with a slight offset from zero, with more higher abundance proteins in AD CSF (Fig R2A). Inspecting the relation of fold-change distribution and general protein abundance revealed that the most abundant proteins had a reduced relative intensity in AD CSF. This even held true in the completely independently measured Higginbotham et al. study (Fig. R2B-E). That study also showed a pronounced asymmetry of significant proteins to be increased rather than decreased in AD (Fig. R2F), in agreement with our results.

Figure R2: AD/non-AD fold change distribution and parameters of intensity normalization in the Sweden cohort.

A) Histogram of AD/non-AD fold changes. The number of proteins with a fold change above or below 0, respectively, are given. Numbers do not add up to 1484 because some proteins were exclusively identified in either the AD or the non-AD samples.

B-E) Interrelation of protein intensity and AD/non-AD fold changes. Red boxes highlight highly abundant proteins. The median protein intensity has been calculated globally across the three cohorts of this study and has also been used as abundance measure for the CSF1 dataset for comparability.

F) AD/non-AD Volcano plot for CSF1 dataset by Higginbotham et al.

Also, since tau levels were a criterion for assignment of patients to the AD groups, why should it be surprising or noteworthy that tau was a highly changing protein between AD and controls?

Confident quantification of tau and its increase in AD CSF provides an important positive control for our relatively new MS-based approach. This is indeed reassuring rather than surprising.

4. page 7, second paragraph: Berlin AD patients were required to have >350 ng/L t-tau, but the other 2 cohorts required >400 ng/L t-tau (Sweden) or > 450 ng/L t-tau (Magdeburg). Could this explain some of the differences in number of differentially expressed proteins? This assumes that t-tau levels correlate with other protein changes- not unlikely given that the protein ratio cutoff is 1.3.

The new AD classification criteria requires > 400 ng/L t-tau for all cohorts and result in a greater AD/non-AD effect size and better agreement with the two other cohorts, see above. Using these new AD criteria, there are only two AD patients in the Berlin cohort with a t-tau concentration between 400 and 450 ng/L. Only two of the 33 AD samples changed status when increasing the t-tau-cutoff from 400 to 450 ng/L and this does not change the proteomics results much (although it is true that levels of other proteins do change with t-tau (Fig 4A-D).

5. Bottom of page 7: It is not at all remarkable that 191 proteins changed in both Magdeburg and Sweden cohorts. If out of 1200 proteins 500 are changed in each, 191 seems to be a virtually random overlap between the 2 groups of 500. It would be much more interesting to see if the percent overlap is much better than approximately 40% (191/approximately 500) if the statistical requirements for change were more stringent than uncorrected p-value of < 0.05. Also, was the direction of change required to be consistent to qualify as overlapping? It should be.

We thank the reviewer for this insight and agree that it is better to focus on the overlap after applying more stringent criteria first, including requiring the same directionality up front as the suggested. We already partially discussed this in the summary above. In the revised manuscript, we first filtered for p-value significantly changing proteins between AD and controls. We then determined the fold-change correlation of the common 172 proteins, which turned out to be high at a Pearson correlation coefficient at 0.91. The corresponding values for the other two comparisons are likewise high at 0.80 and 0.90 (Sweden vs. Berlin and Magdeburg/Kiel vs. Berlin, respectively) (Fig 3C-E). This suggests that there are indeed cohort specific effects, but when focusing on the proteins changing in common between cohorts, the quantitative agreement between them points to true underlying pathological processes. This notion is also supported by the nature of the potential biomarkers in the 40 protein signature, which includes not only tau itself but also proteins like SOD1 and SOD2 as well as PARK7 and YKL-40.

6. page 8, first paragraph: It is worrying that of the 37 proteins differing "significantly" by AD status in all 3 cohorts, only 28 were in the same direction. To me this indicates a problem with the overall dataset.

We agree and note that this number is now reduced to 3 out of 43 with the new, uniform classification. Furthermore, we have now filtered out proteins with opposite directionality up front, as mentioned just above.

Narrowing the panel to tau-associated proteins now biases the proteomic assay to samples that have increased tau, which was a prerequisite for inclusion in the AD groups. The proteomic assay would then simply reinforce the ELISA tau measurements made at the point of diagnosis, rather than indicate AD per se.

We agree that this was somewhat circular, and in the revised manuscript we have dropped this filtering step.

7. Page 8, 3rd paragraph, last sentence: is this about one patient or more? "The second-most AD-like participants... suggesting that they had an AD-like phenotype". Please clarify or correct. The reclassification of the Berlin group based on proteomic signatures implies that the original diagnoses of AD vs control were faulty. By definition the new Berlin groups would behave better in the proteomic assays as shown in Fig 3. But what about the clinical status of the rest of the Berlin cohort in addition to the top 2 most AD-like subjects? Did these statuses agree with the proteomics predictions? Also, would the original classification of the Berlin cohort look more like the reclassified cohort if more than 350 ng/L t-tau have been required?

Indeed, this sentence pertained to one patient. This is not relevant any more due to the uniform classification and this observation is not mentioned in the main text any more. Likewise, the reclassification of the Berlin cohort is now eliminated entirely from the revised manuscript.

8. Page 11, proteins differing by AD status correlate with CSF t-tau abundance: Again, this should be expected since the AD cohorts were largely assigned based on t-tau measurements by ELISA.

We agree with the reviewer that it should be expected that proteins differing by AD status correlate with CSF t-tau abundance. However, we note that this was not necessarily the case in previous proteomics studies and that this further supports the quality of our data set. We hope that our dataset can be used by the community to investigate the strength of association of proteins of interest with t-tau abundance.

9. page 13, bottom: Actually, of 108 proteins, 42 significant proteins in agreement with the Higginbotham study (even fewer with directional agreement, which are the only proteins that should count) does not seem like good agreement- this suggests that the majority of biomarker candidates are not confirmed. Therefore the sentence about high correlation at the top of page 14 should be toned down a bit.

Due to the unified classification, the consistency with the Higginbotham study is drastically higher. Nevertheless, we have toned down the sentence as suggested, which now reads like this in the revision:

“Out of our 40 protein signature 38 proteins were contained in the dataset of this independent study and 26 of 38 (68%) thereof were also significant (Fig EV5A). This is a highly significant enrichment amongst all significant proteins in the dataset of that independent study (odds ratio 10, $p < 0.0001$, Fig EV5B)”

We have also toned down the conclusion sentence (initially at the top of page 14):

“Taken together, AD-associated proteins signatures identified in our work are validated in a completely separate study using an independent cohort and different experimental strategy.”

Would percent agreement between the 2 studies increase if inclusion criteria (lower p-value, or use of q-value or corrected p-value) were more stringent?

The agreement of directionality between studies increased from 31 out of 35 proteins using the cohort-specific AD classification before to 26 of 26 using the uniform classification criteria in the revised manuscript ($p < 0.05$ each). Likewise, indeed, the percentage of proteins significant in our study and contained in the Higginbotham et al. CSF1 dataset that are also significantly changing in that latter dataset increases from 68% to 83% when increasing the stringency of filtering in our study from $p < 0.05$ to $q < 0.05$.

Minor points

10. page 13, 2nd paragraph line 3: should be "differed significantly.."

Thank you. This has been corrected.

Reviewer 2

In this study, Bader et al. performed unbiased proteomic analysis of a total 208 AD and non-AD CSF samples derived from three separate cohorts. These cohorts were notably stratified into non-AD and AD groups in varied ways. While the Sweden cohort exclusively used biochemical CSF levels of Abeta and tau to diagnostically classify their cases, the Magdeburg / Kiel and Berlin cohorts used a combination of clinical presentation and biochemical data. Also notable, the Sweden and Magdeburg / Kiel biochemical measures (t-tau, p-tau181, Abeta1-42) differed from those of the Berlin group (t-tau, Abeta1-40, Abeta1-42).

Using a streamlined automated approach for sample digestion, the 208 samples were analyzed via DIA mass spectrometry, resulting in the quantification of an average 1200 proteins per sample. High inter-assay reproducibility was achieved for over 1000 of the proteins ($CV < 20\%$). The authors then used a clustering analysis to examine functional relationships between the quantified proteins, identifying a relatively large synapse-associated MAPT-containing protein group. Differential expression analysis of quantified proteins revealed robust levels of differential protein expression ($p < 0.05$) between the AD and non-AD groups of the Sweden and

Magdeburg / Kiel cohorts (~400 to 500 proteins). Most of these altered proteins were increased in AD CSF and enriched with neuron-associated gene ontologies. However, the Berlin group featured a markedly lower number of differentially expressed proteins between its AD and non-AD subjects (~100 proteins). The authors linked this result to the poor biochemical (i.e. AD biomarker) separation the Berlin cohort demonstrated in its clinical amyloid and tau assays. MAPT and YWHAG were two of the most significantly increased proteins in AD among all three cohorts.

The authors found consistent protein changes across the Sweden and Magdeburg / Kiel cohorts (~200 proteins). However, when looking for consistent protein changes across all three cohorts, only 37 proteins were identified. The authors then narrowed these 37 proteins to an 11-protein panel by including only those found in the tau-containing cluster of their initial global correlation map. These 11 proteins were then used to re-classify their Berlin cohort into control-like, intermediate, and AD-like clusters. The authors then looked for consistent protein changes across the three cohorts using this re-classified Berlin cohort, excluding the intermediate group. This produced a panel of 130 proteins, which they narrowed to 31 proteins based on consistent changes within an independent CSF analysis performed by Higginbotham et al. Machine learning was then used to further narrow these 31 proteins to 13 CSF protein signatures of AD.

Overall, this study was interesting and highlighted advancements in high-throughput proteomic methods for CSF analysis with regards to proteome depth and reproducibility. The authors should also be commended for comparing their AD biomarker findings with recently deposited pre-prints as this enhances the rigor and reproducibility of their results. However, there are several major limitations with interpretations of the results mainly involving the rubric used for biomarker prioritization that need to be addressed, prior to publication.

We thank the reviewer for the overall positive and insightful evaluation of our work, especially with regards to the interpretation of results. The limitations alluded to mainly pertain to the reclassification and this is addressed in the revision as outlined in the summary above and in detail in the responses to this Reviewer and Reviewer 1 above.

Major points

- First, the authors' rubric for marker prioritization in the Berlin cohort is disturbingly circular. As presented, the Berlin cohort was re-classified based on the discovery proteomic results, essentially forcing it into proteomic consistency with the other two cohorts. Then, consistency across these three cohorts was used as a criterion to prioritize biomarkers following this re-classification. The authors justify this circular rubric using scant anecdotal evidence and suggests that several individuals in this Berlin cohort were in the presymptomatic stages of AD and simply misclassified at the outset. However, should not this presumption be further substantiated?

We agree with this Reviewer and Reviewer 1, but note that this was due to sticking with the classification of the clinical centers themselves. As outlined in the summary above, we have completely eliminated the proteomics-based reclassification of the Berlin cohort and classified all three cohorts consistently based on clinical AD CSF biomarker concentrations.

There are established research guidelines for the classification of presymptomatic AD based on cognitive and core biomarker status. Would it not make more sense-and generate a much less convoluted rubric-to re-stratify the Berlin individuals based on these guidelines? For instance, Tau/Abeta ratios are generally accepted as the most accurate way of biochemically defining cognitive normal individuals with presymptomatic disease. Yet, such ratios were not mentioned in the results of the manuscript.

There is unfortunately no universally accepted classification scheme for AD in clinical practice. Many studies and several meta-analysis report a range of diagnostic performance parameters for single biomarkers or combinations thereof including tau/A β ratios, however, a general consensus on the best integration of multiple AD CSF biomarkers has not been reached (Ritchie *et al*, 2017; Ferreira *et al*, 2014; Bloudek *et al*, 2011). The Hulstaert index integrate the concentrations of t-tau and A β_{1-42} into a ratio, has been reported to perform better diagnostically than a simple ratio of t-tau/A β_{1-42} (Molinuevo *et al*, 2013; Hulstaert *et al*, 1999; Vos *et al*, 2013). This index is calculated as $A\beta_{1-42} / (240 + (1.18 * t\text{-tau}))$ using ng/L concentrations for both core biomarkers and a threshold of 1. Thus we compare the results obtained by the Hulstaert index to the results of our uniform AD classification that is the basis of the revised manuscript.

Using the Hulstaert index the three cohorts show varying degrees of separation of the AD and non-AD groups (Fig R3A). Agreement of the Hulstaert classification with our uniform AD classification was very high for the Sweden and Magdeburg/Kiel cohorts (Fig R3B). In the Berlin cohort, classifications were also largely consistent, however, the Hulstaert index was less stringent for AD with more samples in the grey zone regarding the clinical AD CSF biomarkers classified as AD.

Reassuringly, the proteomics results using the Hulstaert index were similar to the results obtained under our new uniform classification. The Hulstaert three-cohort intersection comprised 22 consistent out of 26 significant proteins (85%) compared to 40 out of 43 (93%) in our uniform classification (Fig R3C). These two sets largely overlapped with 20/22 (91%) also found using the uniform classification (Fig R3D). Taken together, the Hulstaert index for AD classification results in a similar but smaller (22 vs 40) and less consistent (85% vs 93%) panel of AD marker proteins.

We conclude that our uniform classification appears to be more stringent regarding AD assignment by stipulating two cutoffs to be met independently and more refined by also including the A β_{1-40} information.

In the revised manuscript we summarize the above discussion in the Materials and Methods with the new supplementary figure below (Appendix Fig S6).

In the revised discussion we added:

“There is no universally accepted AD classification system, however, various different integration schemes of clinical AD CSF biomarkers have been explored (Ritchie *et al*, 2017; Ferreira *et al*, 2014; Bloudek *et al*, 2011). Using the Hulstaert index, a variation of the A β_{1-42} /t-tau ratio, for AD classification of the three cohorts we obtained largely the same, but fewer

statistically significant potential marker proteins compared to our uniform AD classification (Appendix Fig S6A-D, Materials and Methods) (Hulstaert et al, 1999; Molinuevo et al, 2013; Vos et al, 2013).”

In Material and Methods:

“The Hulstaert index for AD classification is a variant of the $A\beta_{1-42}/t$ -tau ratio with improved diagnostic performance (Molinuevo et al, 2013). It is calculated as $A\beta_{1-42}/(240 + (1.18*t\text{-tau}))$ using ng/L concentrations and samples below a cutoff value of one are classified as AD (Hulstaert et al, 1999). We performed an independent analysis using the Hulstaert index instead of our uniform classification. As shown in Appendix Figure S6, the results overlap almost completely, however, the Hulstaert index, although less stringent in AD inclusion, leads to a smaller number of significantly different proteins.”

Appendix Figure S6. Proteomics results when using the Hulstaert index for AD classification

- Hulstaert indices of AD and non-AD populations of this study
- Agreement between the Hulstaert index-based AD classification and our standard classification as described in Materials and Methods section.

- C) Intersection of significant ($p < 0.05$) proteins across the three cohorts of our study when using the Hulstaert index for AD classification. AD/non-AD fold changes in each cohort shown by the heatmap and proteins ranked according to the mean fold change.
- D) Overlap of significantly ($p < 0.05$) and consistently AD-regulated proteins between the Hulstaert index and our standard AD classification

The criteria used to determine cognitive status was also not clearly defined in the results. It is possible stricter MMSE cut-offs could have better stratified this Berlin group as well, prior to proteomic analysis?

We agree with the Reviewer that classifying the Berlin cohort based on MMSE scores is an interesting avenue to explore, which we have done for the revision of the paper. In the literature, reference mean MMSE scores were 29, 27, and 20 for cognitively normal, mild cognitive impairment (MCI), and AD participants in a large study (Chapman *et al*, 2016). We explored various cutoffs from MMSE>29 to MMSE<21 (Fig EV4B-F). Of our 40 protein AD signature the proteins upregulated in AD (35 proteins) are shown in red and the down-regulated (5 proteins) in blue. As can be seen in the figure the trend is preserved in the MMSE classification of the Berlin cohort at all cutoffs. Thus, MMSE-based and clinical AD CSF biomarker-based classification agreed in the sense that proteins associated with biochemically defined AD also associated with cognitive performance. Amongst the MMSE-score cutoffs tested, a cutoff score of 25 (corresponding to mild cognitive impairment) resulted in the best separation with most proteins significantly different between the high score and the low score group.

It is encouraging that we find largely the same proteomic outliers when following a completely independent patient classification (one following cognitive rather than biochemical indications). Nevertheless, while the MMSE evaluation is widely used for screening due to its simplicity, it has limited diagnostic accuracy (Mitchell, 2009; Perneczky *et al*, 2006; Arevalo-Rodriguez *et al*, 2015). For this reason and to ensure comparability with the other two cohorts in which MMSE scores were not available, we did not use the MMSE scores for AD classification of the Berlin cohort.

This analysis is now included in the in the Results section:

“When stratifying the Berlin cohort into ‘high MMSE score’ and ‘low MMSE score’ groups over a range of 29 to 21, we obtained the greatest separation at a cutoff of 25. Reassuringly, MAPT and YWHAG were the top outliers and our 40 protein signature showed the expected association with the MMSE groups at all cutoff values in spite of the limited diagnostic performance of the MMSE evaluation (Fig EV4B-F) (Mitchell, 2009; Perneczky *et al*, 2006; Arevalo-Rodriguez *et al*, 2015). Thus, CSF protein signatures linked to biochemically defined AD also associate with cognitive performance.”

Expanded View Figure 4. MMSE score correlation analysis and proteome alterations when stratifying the Berlin cohort by MMSE score

A) Correlation of proteins to the mini mental state examination (MMSE) scores in the Berlin cohort. Proteins with a correlation q-value below 0.05 are labeled. Proteins of the 40 protein signature are colored in red for proteins with increased abundance in AD CSF and in blue for proteins with increased abundance in non-AD CSF.

B-F) CSF proteome alterations between groups of lower MMSE-scores (poor neuropsychological performance) and groups of higher MMSE-score as separated by cutoffs of 29 (A), 27 (B), 25 (C), 23 (D), and 21 (E). Proteins of the 40 protein signature are colored in red for proteins with increased abundance in AD CSF and in blue for proteins with increased abundance in non-AD CSF.

In summary, the authors should eliminate the circular proteomic re-classification of the Berlin group and explore defining them based on more rigorous clinical and core biomarker research criteria used in the field or exclude the cohort entirely as the re-classification confounds the AD biomarker findings.

•A second concern generated by the proteomic re-classification of the Berlin group is the newly created presence of an "intermediate" sub-group. This group included a third of the total Berlin cohort, including 15 individuals originally classified as AD. This intermediate group was promptly identified and eliminated from analysis in the second half of the manuscript. Yet, there was no discussion as to what these "intermediate" individuals signify. Do the authors consider all these cases presymptomatic AD? Or are they more consistent with MCI or mild AD? Could a portion of these intermediate individuals represent a degree of biological heterogeneity among true AD cases? This is simply another unclear / unsatisfying aspect of the circular re-classification of the Berlin group.

We completely agree and have done this in uniform manner for all the analyses in the revised paper. As outlined in the summary, the Berlin cohort does add substantially to our conclusions when that classification is used. The fact that our potential biomarker panel is preserved in three cohorts (and even a fourth one, when adding the Higginbotham et al. study), in our view makes it much more reliable.

•Third, perhaps due to the issues outlined above, the overall scheme of the paper was confusing and difficult to follow. When the authors mentioned a new protein panel of interest, I often had to revisit previous paragraphs to determine the panel's derivation. Could the process of marker prioritization perhaps be more streamlined? A global depiction of the algorithm would be helpful.

Thank you for this feedback. We have reduced the number of steps in the bioinformatics data analysis workflow, e.g. there is no selection for tau-correlating proteins according to the global correlation map and no Berlin reclassification in the revised manuscript. We have also included an updated version of the flow chart narrowing down the proteins (Fig EV5A). Moreover, we define one important set of proteins as '40 protein signature' early in the manuscript and refer to this panel throughout the remainder of the manuscript:

“We assessed whether AD and non-AD samples clustered together independent of the cohort, based on either the global unfiltered CSF proteome profile, the less stringent ($p < 0.05$) intersection, or the more stringent ($q < 0.05$) intersection set of proteins significant in all three cohorts. After Z-scoring protein intensities within cohorts, unsupervised clustering clearly separated AD from non-AD groups in all three cases (global proteome, both intersection sets) (Fig 3F-G, Appendix Fig S2A,B). In the $p < 0.05$ intersection set 40 out of 43 proteins (93%) differed consistently in abundance by AD status, 35 of which had an elevated abundance in AD CSF and five an elevated abundance in non-AD CSF (Fig 3F, Appendix Fig S3A,B). We discuss these proteins as ‘the 40 protein signature’ of AD in the remainder of this paper.”

We likewise streamlined the machine learning part as a result of the consisted patient classification. We added a more consistent description about the feature selection:

All classifiers reached an area under the ROC curve (AUC) of at least 0.84. XGBoost had the best performance with a mean AUC of 0.91 and was selected for further analysis. To determine the optimal number of features, we iteratively added them in their order of importance in the decision tree. The overall model performance increased with the number of proteins and reached a plateau at six proteins (MAPT, PKM [P14618-2 isoform], MIF, IMPA1, YWHAZ, ALDOC), which we selected for the final model.

Minor points

•Pg 4. The following sentence needs to be reworded, as it is awkward and the numbers do not compute: "Accordingly, we collected and analyzed three separate study populations each comprised around 30 AD subjects and 30 non-AD controls, totaling 208 individuals."

The sentence has been changed to:

"To implement the rectangular strategy, we analyzed three separate study populations of about 30 AD patients and about 30 or 50 controls, amounting to 197 individuals in total."

•Pg 4. The precise clinical and biochemical cut-offs to classify non-AD and AD individuals in the three cohorts should be explicitly stated.

The precise biochemical thresholds used for this study are stated in the revised Materials and Methods section as follows:

"Patients were classified as AD if the t-tau concentration was above 400 ng/L and the $A\beta_{1-42}$ concentration below 550 ng/L or the $A\beta_{1-42}/A\beta_{1-40}$ ratio was below 0.065. The t-tau criterion and at least one of the two $A\beta$ criteria had to be met for a patient to be classified as having AD and patients were classified as not having AD otherwise."

•Pg. 4. The non-AD sub-group of the Magdeburg/Keil cohort was substantially younger than the AD group. What effect does this have on the proteomic results? Were age and sex regressed as co-variates in the bioinformatic workflow? The authors should discuss and justify.

Thank you for pointing this out. We agree with the Reviewer that age and sex should be explored as covariates on top of the general age and sex matching of the cohorts. Following the Reviewer's suggestion, we have added regression models for AD status and t-tau CSF concentrations, each using sex and age as covariates. The results of the regressions reveal age and sex dependencies for several proteins (Fig EV3C, Appendix Fig S4B). However, they still confirm the results of the study regarding the AD-association and correlation with CSF t-tau concentration for the entire 40 protein signature of significantly and consistently AD-regulated proteins.

We have added the following sentences in the Results section:

Association with AD status:

"To specifically investigate the effect of age and sex on the AD-regulation of the 40 protein signature, we employed a linear regression model. After correction for age and sex in this way, the CSF abundance of all 40 proteins still significantly depend on AD status (Fig EV3C).

Interestingly, CSF proteome alterations were of smaller magnitude in males compared to females in this study population.”

Association with CSF t-tau concentration with or without age and sex as covariates:

“We next asked how our 40 protein signature correlated with clinical t-tau measurements. Indeed a large fraction – 29 of 40 proteins – significantly correlated with t-tau in each of the three cohorts, and the directionality of change was also as expected for the non-significant proteins (Fig 4A-E, Appendix Fig S4A). This is a substantial enrichment over the numbers expected by chance in this dataset ($p < 0.0001$, odds ratios 37). Upon adjustment for age, sex, and cohort in a linear regression model comprising all three cohorts, all 40 proteins were significantly associated with t-tau (Material and Methods) (Appendix Fig S4B).”

In the Materials and Methods section we have added:

“We used linear regression analysis computed in RStudio version 1.2.5033 using R version 3.6.3 and assessed the association of log₁₀-transformed protein intensities first with AD status (Fig EV3C) and second with log₁₀-transformed ELISA measured CSF t-tau concentration (Appendix Fig S4B), adjusting for age, sex, and cohort (Sweden, Magdeburg/Kiel, or Berlin) in both models. To compare estimators of binary (AD status, sex) to those of continuous variables (age, t-tau concentration [log₁₀]), the estimators for continuous variables (i.e. per 1 year [age] and per 1 unit in log₁₀ space of t-tau concentration/[ng/L]) were multiplied with the inter-quartile range (IQR) of the variable for plotting. IQRs for age were eleven years for the complete dataset (Fig EV3C) and nine years for the reduced dataset excluding the Kiel samples due to missing t-tau concentration values (Appendix Fig S4B). The t-tau concentration 75% and 25% quantiles were 802ng/L and 275ng/L, respectively, corresponding to an interquartile range of 527 and 0.4648 in linear and log₁₀ space, respectively (Appendix Fig S4B). Regression coefficients for age and sex were displayed in the heat map if the p-value for these estimators was below 0.05. All proteins were associated with AD status or t-tau concentration at a significance below of 0.05 in each plot.”

•Pg 5. The authors state "CSF has much less protein than plasma". This statement should be followed by a citation. Also, how much less protein? Has this been quantified? Was more volume of CSF loaded on the instruments to get equivalent signal?

CSF contains about 200-fold less protein than plasma. References have been inserted. More CSF (40µl) than plasma (1µl) was used for sample preparation. 500 ng of purified peptide were loaded onto the chromatography column for LC-MS/MS analysis in either case.

We have inserted a references for the protein concentrations in the Results section of the revised manuscript:

“CSF contains much less protein than plasma, with about 0.17 – 0.70 g/L and 60 – 80 g/L total protein content, respectively (Seyfert *et al*, 2002; Laub *et al*, 2010; Marshall, 2018).”

The Materials and Methods section contains the volume information for CSF and references the plasma workflow publication:

“The sample preparation was optimized for CSF on the basis of our Plasma Proteome Profiling workflow (Geyer *et al*, 2016). CSF was aliquoted in 96-well plates and processed with an

automated set-up on an Agilent Bravo liquid handling platform. In total 40 µl of CSF were mixed with 40 µl PreOmics lysis buffer (PreOmics GmbH) for reduction of disulfide bridges, cysteine alkylation and protein denaturation at 95°C for 10 min.”

Pg. 5. The authors highlighted the second-largest cluster of their original unbiased global correlation map and used it later in the study to prioritize markers of interest.

We agree with Reviewer 1 and this Reviewer’s comment further below that AD markers should not be narrowed down by selecting proteins co-clustering with tau in the global correlation map. Thus we have removed this filtering step in the revised manuscript.

They in part justify this stating that due to its functional annotations (neuronal, axonal, and synaptic), this cluster reflects "the brain contribution to the CSF proteome". However, this statement is somewhat presumptuous. Is this cluster more enriched than the others with proteins detectable in the brain? Do the other clusters not contain any proteins detectable in brain?

Neuronal or brain-specific annotation terms were selectively enriched in this second largest cluster. By “brain contribution to the CSF” we wanted to state that identification of neuronal proteins in the CSF highlights that proteins originating in the central nervous system (CNS) accumulate in the CSF, thus making the CSF reflective of physiological or pathological proteome alteration in the CNS.

We have changed the relevant sentences in the revised manuscript:

“The global protein correlation map (Wewer Albrechtsen *et al*, 2018) resulting from more than a million protein-protein comparisons highlighted eight main clusters of proteins which follow common functions or themes. For instance, neuronal annotation terms such as the gene ontology cellular compartments (GOCC) terms neuron projection, axon and synapse were selectively enriched in the second largest cluster (Fig 1E, Fig EV2D). Identification of neuronal proteins in the CSF highlights that proteins originating in the central nervous system accumulate in the CSF, thus making the CSF reflective of physiological or pathological proteome alteration in this organ.”

The authors also imply that the other clusters represent some sort of contamination or that their presence is related to "quality issues". However, where is the justification of this? The pre-print paper they reference (Higginbotham *et al.*) demonstrated consistent disease-related CSF changes in plasma / vascular-derived proteins. How is this to be explained if such clusters are only related to technical issues?

We agree with the reviewer that the global correlation map alone does not imply that a given set of proteins, e.g. highly abundant plasma proteins, must be contaminants to the CSF. We have also inserted the Higginbotham *et al.* reference and highlighted that vascular proteins as termed there have been reported to be altered in AD CSF.

However, we have extensive experience with the analysis of plasma proteomes, in which this issue also occurs. In a previous publication, we have shown that one can separate technical issues from true associations based to correlations between sets or clusters of proteins (Geyer *et al*, 2019). Applied to this case, proteins co-clustering with known blood proteins in the CSF likely

originate from blood contamination of CSF, which can arise during sample taking which is hard to avoid entirely. When they are contaminants such groups of proteins are both ‘regulated’ together and also occur in similar abundance relationships to each other as in blood. Conversely, if a protein also found in blood does not correlate with the blood proteins, it may still be a genuine biomarker for AD. The global correlation map presents an efficient approach to distinguish biomarkers from contaminants. Regarding our study we report that it was reassuring that CSF signatures of biologically relevant to AD clearly separated from protein clusters that are at higher risk to be contamination-associated.

In the revised manuscript the relevant sentences directly following the section above has been changed:

“Another cluster was enriched in blood plasma proteins relating to humoral immunity, the complement system or coagulation. Vascular proteins have been reported to be increased in AD brains while decreased in AD CSF (Higginbotham *et al*, 2019). However, apart from disease-associated effects such as a modulation of the blood-brain-barrier, apparent alterations of blood protein abundances in CSF may be caused by blood contamination during CSF sampling which is hard to avoid entirely. Proteins are likely blood contaminants in CSF if they exhibit the same abundance profile across samples as known blood proteins and occur in the same abundance ratio to these blood proteins in CSF as in blood. Conversely, if a protein also found in blood does not correlate with the blood proteins, it may still be a genuine biomarker for AD. The global correlation map presents an efficient approach to distinguish biomarkers from contaminants (Geyer *et al*, 2019). Here, CSF signatures of proteins biologically relevant to AD clearly separated from protein clusters that are at higher risk to be contamination-associated (Fig 1E).”

•Pg 6. The authors state that "...we and others had altogether failed to detect tau or at significantly changed levels in proteomic studies". This is not true, as the pre-print paper the authors referenced (Higginbotham *et al.*) detected changes in tau in CSF1 as well as the published findings from the Pandey group (PMID: 30578620). Several PRM (not dissimilar to DIA-MS) studies have also detected and quantified Tau (J. Proteome Res. 2016, 15, 2, 667-676).

We have referenced these past achievements and inserted all three references in the revised manuscript:

“The fact that tau levels are elevated in AD CSF has been known for more than two decades but this important protein is not easily quantified in large proteomics discovery cohorts. Typically, tau quantitation by mass spectrometry has required extensive fractionation and depletion of abundant proteins, limiting throughput (Sathe *et al*, 2019; Higginbotham *et al*, 2019). Alternatively, targeting instead of discovery strategies can in principle quantify proteins such as tau in larger sample numbers (Barthélemy *et al*, 2016).”

•Pg 12. The authors state that "glycolytic proteins may originate from astrocytes...". The authors should cite the pre-print they cite in their abstract (Johnson *et al.*) as this paper links the metabolic/glycolytic biomarkers in CSF to differences in these same markers in AD brain.

We are not sure what the reviewer means by (Johnson *et al.*). We have inserted the citation of the pre-print as follows:

“Glycolytic proteins may originate from astrocytes as glycolysis in the brain is mainly performed by these cells to provide lactate for oxidative phosphorylation in neurons (Bélanger *et al*, 2011; Higginbotham *et al*, 2019).”

•Pg. 13. The authors should revisit and download the protein fold change information for CSF1 in BioRxiv pre-print by Higginbotham et al. My review of this data shows an increase their number of validated proteins increases from 31 to 38. For instance, the fold changes for CHI3L1, SOD2, and STMN1 are all increased in AD, whereas SERPINF2, KLKB1, FETUB, and APOA2 are all decreased, further validating the authors' results.

We thank the reviewer for pointing this out. After we had downloaded the data from bioRxiv, an updated second version of the manuscript including supplemental data appeared. We have used the updated data for the bioinformatics analysis of the revised manuscript.

•P. 5. "1484 proteins much higher than reported in published literature." This is not correct. There are a number of studies with >2000 proteins identified in CSF by MS (see Pandey publication and the Higginbotham preprint above). Maybe the authors are referring to the number of proteins contributing to protein quantified in CSF without albumin depletion by DIA-MS or the number quantified at a given missingness threshold? It's not clear from this sentence.

We have inserted the references for the Pandey publication and the Higginbotham pre-print and clarified that we refer to the proteome depth achievable by single shot MS without depletion or fractionation.

The relevant section in the revised manuscript:

“To achieve such CSF proteome depth, extensive fractionation and depletion of abundant proteins often combined with isobaric labelling were previously required, with its associated disadvantages (Sathe *et al*, 2019; Higginbotham *et al*, 2019). For a single shot CSF proteomics workflow that is amenable to high throughput and large cohorts this presents an unprecedented depth at high data completeness. “

•The clustering section is highly confusing. If clustering is performed using the 37 proteins that are significantly different between AD and controls among all three cohorts, wouldn't one expect that AD cases in each cohort would cluster together?

We thank the Reviewer for this feedback. The Berlin cohort reclassification has been removed and the clustering paragraph simplified. AD cases do cluster together as expected after Z-scoring of proteins within cohorts (Fig 3F,G; Appendix Fig S2B).

And why did only 27 of the 37 have higher abundance across the 3 cohorts after this analysis?

Using the uniform AD classification 43 proteins are now significant ($p < 0.05$) across the three cohorts. Of these, 40 exhibit consistent directionality of fold changes (up or down).

Also, they could have used these 27 or 37 proteins to re-classify the Berlin cohort. Why go down to the 11 in the tau cluster? Justification is not provided.

We agree with Reviewer 1 and 2 and have removed the step of narrowing proteins down by selecting proteins co-clustering with tau.

- The authors shouldn't speculate that the abundance calculations in brain equate to protein levels in CSF and reflect "leakage" or tissue damage. That may be the case for some proteins, but not all. There is likely exosome release and other mechanisms underlying the levels in brain and CSF.

We agree with the Reviewer that other mechanisms may be responsible for the accumulation of proteins in the CSF. In the revised manuscript, we have toned down our interpretation. We list damage-associated loss of membrane integrity as one possible mechanism of accumulation of abundant proteins in the CSF, alongside exosome release and potentially other mechanisms:

“When we mapped the upregulated proteins in our AD CSF panel onto a deep human brain proteome (Carlyle *et al*, 2017), their abundance in brain was generally in the more abundant range (Fig 4G,H). This observation is consistent with mechanisms in which cellular proteins are released into the CSF by tissue damage-associated loss of membrane integrity, exosome release or others.”

- Fig. 4D, legend states "inversely correlating" with tau. Looks like regular correlation.

We have removed the word “inversely”. Proteins with positive and negative correlation are depicted.

- P. 13, referring to glycolytic signature, the authors should cite Johnson *et al*. pre-print and other previous studies in the discussion when referring to glycolytic proteins elevated in CSF.

We do not understand the reference Johnson *et al*. and assume that the reviewer refers to Higginbotham *et al*. This pre-print and other previous studies have been inserted in the discussion in the revised manuscript:

“Moreover, we identify a number of glucose metabolism-associated proteins elevated in AD CSF in line with other reports (Higginbotham *et al*, 2019; Sathe *et al*, 2019; Dayon *et al*, 2018).”

- Expanded View Fig 2 C-E legend labels are incorrect

The labels have been corrected.

- Supplemental Tables are not well formatted and difficult read.

We have improved the formatting, split complex tables into two smaller ones, and added a better description to the tables.

Reviewer 3

This work is extremely important work and we need new biomarkers and hypotheses on why we get AD. However, this is half finished. The findings needs to be validated by another technique, the days have gone where you run a whole of samples and come to a conclusion based on that because omics often kicks up red herrings. These red herrings can send researchers down a blind ally. Also the reason why there is centre to centre differences haven't been addressed properly.

We thank Reviewer 3 for acknowledging the importance of this work.

Reviewer 3 further highlights the reproducibility issues faced by previous biomarker discovery studies utilizing omics-approaches and for that reason suggests to validate the results by an orthogonal technique.

In this regard, we consider the results of the Higginbotham et al. study a validation by a completely independent laboratory, using quite different technologies for proteome analysis. In our view, this amounts to as good a validation as one could hope for with any other technique. With regards to the studies, it is notable that this is the fourth independent cohort. Although in principle ELISA assays could be developed for each of the 40 proteins in our signature, this would be beyond the scope of the current study. Even when available, performing 40 ELISAs with stringent quality control on nearly 200 clinical samples would not be easy at all. We conclude that four independent cohorts provide strong enough evidence not requiring further experiments by affinity-based assays.

References:

- Arevalo-Rodriguez I, Smailagic N, Roque I Figuls M, Ciapponi A, Sanchez-Perez E, Giannakou A, Pedraza OL, Bonfill Cosp X & Cullum S (2015) Mini-Mental State Examination (MMSE) for the detection of Alzheimer's disease and other dementias in people with mild cognitive impairment (MCI). *Cochrane database Syst. Rev.*: CD010783
- Barthélemy NR, Fenaille F, Hirtz C, Sergeant N, Schraen-Maschke S, Vialaret J, Buée L, Gabelle A, Junot C, Lehmann S & Becher F (2016) Tau Protein Quantification in Human Cerebrospinal Fluid by Targeted Mass Spectrometry at High Sequence Coverage Provides Insights into Its Primary Structure Heterogeneity. *J. Proteome Res.* **15**: 667–676 Available at: <https://doi.org/10.1021/acs.jproteome.5b01001>
- Bélangier M, Allaman I & Magistretti PJ (2011) Brain energy metabolism: Focus on Astrocyte-neuron metabolic cooperation. *Cell Metab.* **14**: 724–738
- Bloudek LM, Spackman DE, Blankenburg M & Sullivan SD (2011) Review and meta-analysis of biomarkers and diagnostic imaging in Alzheimer's disease. *J. Alzheimers. Dis.* **26**: 627–645
- Callister SJ, Barry RC, Adkins JN, Johnson ET, Qian W-J, Webb-Robertson B-JM, Smith RD & Lipton MS (2006) Normalization approaches for removing systematic biases associated with mass spectrometry and label-free proteomics. *J. Proteome Res.* **5**: 277–286
- Carlyle BC, Kitchen RR, Kanyo JE, Voss EZ, Pletikos M, Sousa AMM, Lam TKT, Gerstein

- MB, Sestan N & Nairn AC (2017) A multiregional proteomic survey of the postnatal human brain. *Nat. Neurosci.* **20**: 1787–1795 Available at: <http://dx.doi.org/10.1038/s41593-017-0011-2>
- Chapman KR, Bing-Canar H, Alosco ML, Steinberg EG, Martin B, Chaisson C, Kowall N, Tripodis Y & Stern RA (2016) Mini Mental State Examination and Logical Memory scores for entry into Alzheimer’s disease trials. *Alzheimers. Res. Ther.* **8**: 9
- Dayon L, Núñez Galindo A, Wojcik J, Cominetti O, Corthésy J, Oikonomidi A, Henry H, Kussmann M, Migliavacca E, Severin I, Bowman GL & Popp J (2018) Alzheimer disease pathology and the cerebrospinal fluid proteome. *Alzheimer’s Res. Ther.* **10**: 1–12
- Ferreira D, Perestelo-Pérez L, Westman E, Wahlund LO, Sarrisa A & Serrano-Aguilar P (2014) Meta-review of CSF core biomarkers in Alzheimer’s disease: The state-of-the-art after the new revised diagnostic criteria. *Front. Aging Neurosci.* **6**: 1–24
- Geyer PE, Kulak NA, Pichler G, Holdt LM, Teupser D & Mann M (2016) Plasma Proteome Profiling to Assess Human Health and Disease. *Cell Syst.* **2**: 185–195 Available at: <http://dx.doi.org/10.1016/j.cels.2016.02.015>
- Geyer PE, Voytik E, Treit P V, Doll S, Kleinhempel A, Niu L, Muller JB, Buchholtz M-L, Bader JM, Teupser D, Holdt LM & Mann M (2019) Plasma Proteome Profiling to detect and avoid sample-related biases in biomarker studies. *EMBO Mol. Med.* **11**: e10427
- Higginbotham L, Ping L, Dammer EB, Duong DM, Zhou M, Gearing M, Johnson ECB, Hajjar I, Lah JJ, Levey AI & Seyfried NT (2019) Integrated Proteomics Reveals Brain-Based Cerebrospinal Fluid Biomarkers in Asymptomatic and Symptomatic Alzheimer’s Disease. *bioRxiv*: 806752 Available at: <http://biorxiv.org/content/early/2019/10/16/806752.abstract>
- Hosp F, Gutierrez-Angel S, Schaefer MH, Cox J, Meissner F, Hipp MS, Hartl F-U, Klein R, Dudanova I & Mann M (2017) Spatiotemporal Proteomic Profiling of Huntington’s Disease Inclusions Reveals Widespread Loss of Protein Function. *Cell Rep.* **21**: 2291–2303
- Hulstaert F, Blennow K, Ivanoiu A, Schoonderwaldt HC, Riemenschneider M, De Deyn PP, Bancher C, Cras P, Wiltfang J, Mehta PD, Iqbal K, Pottel H, Vanmechelen E & Vanderstichele H (1999) Improved discrimination of AD patients using beta-amyloid(1-42) and tau levels in CSF. *Neurology* **52**: 1555–1562
- Laub R, Baurin S, Timmerman D, Branckaert T & Strengers P (2010) Specific protein content of pools of plasma for fractionation from different sources: impact of frequency of donations. *Vox Sang.* **99**: 220–231 Available at: <https://pubmed.ncbi.nlm.nih.gov/20840337>
- Marshall W (2018) Total protein. *Assoc. Clin. Biochem. Lab. Med. Anal. Monogr. alongside Natl. Lab. Med. Cat.* Available at: <http://www.acb.org.uk/whatwedo/science/AMALC.aspx> [Accessed March 17, 2020]
- Mitchell AJ (2009) A meta-analysis of the accuracy of the mini-mental state examination in the detection of dementia and mild cognitive impairment. *J. Psychiatr. Res.* **43**: 411–431
- Molinuevo JL, Gispert JD, Dubois B, Heneka MT, Lleo A, Engelborghs S, Pujol J, De Souza LC, Alcolea D, Jessen F, Sarazin M, Lamari F, Balasa M, Antonell A & Rami L (2013) The AD-CSF-index discriminates alzheimer’s disease patients from healthy controls: A validation study. *J. Alzheimer’s Dis.* **36**: 67–77

- Perneczky R, Wagenpfeil S, Komossa K, Grimmer T, Diehl J & Kurz A (2006) Mapping scores onto stages: mini-mental state examination and clinical dementia rating. *Am. J. Geriatr. Psychiatry* **14**: 139–144
- Ritchie C, Smailagic N, Noel-Storr AH, Ukoumunne O, Ladds EC & Martin S (2017) CSF tau and the CSF tau/ABeta ratio for the diagnosis of Alzheimer's disease dementia and other dementias in people with mild cognitive impairment (MCI). *Cochrane database Syst. Rev.* **3**: CD010803–CD010803 Available at: <https://pubmed.ncbi.nlm.nih.gov/28328043>
- Sathe G, Na CH, Renuse S, Madugundu AK, Albert M, Moghekar A & Pandey A (2019) Quantitative Proteomic Profiling of Cerebrospinal Fluid to Identify Candidate Biomarkers for Alzheimer's Disease. *Proteomics. Clin. Appl.* **13**: e1800105–e1800105 Available at: <https://pubmed.ncbi.nlm.nih.gov/30578620>
- Seyfert S, Kunzmann V, Schwertfeger N, Koch HC & Faulstich A (2002) Determinants of lumbar CSF protein concentration. *J. Neurol.* **249**: 1021–1026
- Vos SJB, van Rossum IA, Verhey F, Knol DL, Soininen H, Wahlund L-O, Hampel H, Tsolaki M, Minthon L, Frisoni GB, Froelich L, Nobili F, van der Flier W, Blennow K, Wolz R, Scheltens P & Visser PJ (2013) Prediction of Alzheimer disease in subjects with amnesic and nonamnesic MCI. *Neurology* **80**: 1124–1132
- Wewer Albrechtsen NJ, Geyer PE, Doll S, Treit P V, Bojsen-Moller KN, Martinussen C, Jorgensen NB, Torekov SS, Meier F, Niu L, Santos A, Keilhauer EC, Holst JJ, Madsbad S & Mann M (2018) Plasma Proteome Profiling Reveals Dynamics of Inflammatory and Lipid Homeostasis Markers after Roux-En-Y Gastric Bypass Surgery. *Cell Syst.* **7**: 601-612.e3

23rd Apr 2020

Manuscript Number: MSB-19-9356R

Title: Proteome Profiling in Cerebrospinal Fluid Reveals Novel Biomarkers of Alzheimer's Disease

Thank you for sending us your revised manuscript. We have now heard back from the two reviewers who were asked to evaluate your study. As you will see below, both reviewers acknowledge that you have done an excellent job in addressing their concerns and they are now supportive of publication.

Before we formally accept the manuscript for publication, we would ask you to address a few remaining editorial issues listed below:

- Our data editors have noticed some unclear or missing information in the figure legends, (please see the attached .doc file). Please make all requested text changes using the attached file and *keeping the "track changes" mode* so that we can easily access the edits made.

- Please remove the "available at + link" from the reference list.

- Please include callouts to Datasets EV1-EV7 in the main text.

- Please provide a "standfirst text" summarizing the study in one or two sentences (approximately 250 characters) and three to four "bullet points" highlighting the main findings.

Please resubmit your revised manuscript online, with a covering letter listing amendments and responses to each point raised by the referees. Please resubmit the paper ****within one month**** and ideally as soon as possible. If we do not receive the revised manuscript within this time period, the file might be closed and any subsequent resubmission would be treated as a new manuscript. Please use the Manuscript Number (above) in all correspondence.

Click on the link below to submit your revised paper.

Link Not Available

If you do choose to resubmit, please click on the link below to submit the revision online before 23rd May 2020.

Link Not Available

IMPORTANT: When you send your revision, we will require the following items:

1. the manuscript text in LaTeX, RTF or MS Word format
2. a letter with a detailed description of the changes made in response to the referees. Please specify clearly the exact places in the text (pages and paragraphs) where each change has been made in response to each specific comment given
3. three to four 'bullet points' highlighting the main findings of your study
4. a short 'blurb' text summarizing in two sentences the study (max. 250 characters)
5. a 'thumbnail image' (550px width and max 400px height, Illustrator, PowerPoint or jpeg format), which can be used as 'visual title' for the synopsis section of your paper.
6. Please include an author contributions statement after the Acknowledgements section (see <https://www.embopress.org/page/journal/17444292/authorguide#manuscriptpreparation>)
7. Please complete the CHECKLIST available at (<http://bit.ly/EMBOPressAuthorChecklist>). Please note that the Author Checklist will be published alongside the paper as part of the transparent process (<https://www.embopress.org/page/journal/17444292/authorguide#transparentprocess>).
8. Please note that corresponding authors are required to supply an ORCID ID for their name upon submission of a revised manuscript (EMBO Press signed a joint statement to encourage ORCID adoption) (<https://www.embopress.org/page/journal/17444292/authorguide#editorialprocess>).

Currently, our records indicate that the ORCID for your account is 0000-0003-1292-4799.

Link Not Available

The system will prompt you to fill in your funding and payment information. This will allow Wiley to send you a quote for the article processing charge (APC) in case of acceptance. This quote takes into account any reduction or fee waivers that you may be eligible for. Authors do not need to pay any fees before their manuscript is accepted and transferred to the publisher.

*** PLEASE NOTE *** As part of the EMBO Press transparent editorial process initiative (see our Editorial at <http://dx.doi.org/10.1038/msb.2010.72>), Molecular Systems Biology will publish online a Review Process File to accompany accepted manuscripts. When preparing your letter of response, please be aware that in the event of acceptance, your cover letter/point-by-point document will be included as part of this File, which will be available to the scientific community. More information about this initiative is available in our Instructions to Authors. If you have any questions about this

initiative, please contact the editorial office (msb@embo.org).

Reviewer #1:

The authors have done an excellent job of responding to my previous concerns. Reclassification of the AD vs. control samples using uniform biochemical criteria for all 3 cohorts was an effective solution to the serious problem of comparing disparate patient groups as was done in the first submission. This revised experimental design required a thorough reanalysis of the data, which now make a great deal more sense and lead to a number of now supportable conclusions. The resulting manuscript is now a strong demonstration of the authors' biomarker discovery/validation platform, and may provide a useful proteomic biomarker dataset for the AD research community.

Reviewer #2:

Overall the authors did an outstanding job addressing the reviewer concerns. I only ask for a minor revision related to my previous request to cite the Johnson et al paper when the authors connect their CSF findings to glycolysis (on pg. 12 and pg. 18).

See publication: Johnson, E.C.B., Dammer, E.B., Duong, D.M. et al. Nat Med (2020).
<https://doi.org/10.1038/s41591-020-0815-6>

Again, these are important and consistent findings that the authors have made to the field.

Corresponding Author Name: Matthias Mann
Journal Submitted to: Molecular Systems Biology
Manuscript Number: MSB-19-9356